# Evaluation and Analysis of AMSR2 and FY3B Soil Moisture Products by an In Situ Network in Cropland on Pixel Scale in the Northeast of China

**Haoyang Fu** [1,2]**, Tingting Zhou** [1] **and Chenglin Sun** [1,*]

1   Coherent Light and Atomic and Molecular Spectroscopy Laboratory, College of Physics, Jilin University, Changchun 130012, China; fuhy16@mails.jlu.edu.cn (H.F.); ttzhou18@mails.jlu.edu.cn (T.Z.)
2   College of Electronic Science and Engineering, Jilin University, Changchun 130012, China
*   Correspondence: chenglin@jlu.edu.cn; Tel.: +86-150-4307-6988

**Abstract:** An in situ soil moisture observation network at pixel scale is constructed in cropland in the northeast of China for accurate regional soil moisture evaluations of satellite products. The soil moisture products are based on the Japan Aerospace Exploration Agency (JAXA) algorithm and the Land Parameter Retrieval Model (LPRM) from the Advanced Microwave Scanning Radiometer 2 (AMSR2), and the products from the FengYun-3B (FY3B) satellite are evaluated using synchronous in situ data collected by the EC-5 sensors at the surface in a typical cropland in the northeast of China during the crop-growing season from May to September 2017. The results show that the JAXA product provides an underestimation with a bias ($b$) of -0.094 $cm^3/cm^3$, and the LPRM soil moisture product generates an overestimation with a $b$ of 0.156 $cm^3/cm^3$. However the LPRM product shows a better correlation with the in situ data, especially in the early experimental period when the correlation coefficient is 0.654, which means only the JAXA product in the early stage, with an unbiased root mean square error (ubRMSE) of 0.049 $cm^3/cm^3$ and a $b$ of -0.043 $cm^3/cm^3$, reaches the goal accuracy ($\pm0.05$ $cm^3/cm^3$). The FY3B has consistently obtained microwave brightness temperature data, but its soil moisture product data in the study area is seriously missing during most of the experimental period. However, it recovers in the later period and is closer to the in situ data than the JAXA and LPRM products. The three products show totally different trends with vegetation cover, soil temperature, and actual soil moisture itself in different time periods. The LPRM product is more sensitive and correlated with the in situ data, and is less susceptible to interferences. The JAXA is numerically closer to the in situ data, but the results are still affected by temperature. Both will decrease in accuracy as the actual soil moisture increases. The FY3B seems to perform better at the end of the whole period after data recovery.

**Keywords:** regional soil moisture; in situ network; AMSR2; FY3B; evaluation; EVI; SST

## 1. Introduction

Soil moisture is vital to the earth's water cycle, energy cycle, ecological environment, and agriculture. It is a critical boundary between the land surface and the atmosphere and a key medium surface evapotranspiration [1–7]. Satellite microwave remote sensing technology can be used to monitor surface soil moisture changes in near real time at regional and global scales. Therefore, the evaluation of the accuracy of onboard soil moisture products is of great significance for the calibration of products and the future scientific research on the global water cycle.

In recent decades, satellite remote sensing technology has been continuously developed, and many satellites have been used to monitor various parameters of surface soil. Compared with the

visible light band, microwave remote sensing has long wavelength, strong penetrability, and is not affected by cloud layer and weather conditions. It can realize global all-weather monitoring and ground observation, and is widely used for retrieval of surface soil moisture [8–11] and temperature [12].

In recent years, the L-band is considered to be the most suitable band for soil moisture observation because of its longer wavelength and deeper penetration depth. The Soil Moisture and Sea Salinity (SMOS) mission of the European Space Agency (ESA) can achieve the soil moisture observation at multi-angles [13]. The Soil Moisture Active and Passive (SMAP) mission of the National Aeronautics and Space Administration (NASA) is equipped with a RADAR (stopped transmitting on 7 July 2015) and a radiometer, and it could improve the retrieval accuracy and spatial resolution [14]. Compared to the L-band, the X-band has a much longer temporal sequence of soil moisture observations. The AMSR2 was mounted on the Global Change Observation Mission 1-Water (GCOM-W1) satellite launched on 18 May 2012 and started to acquire observed data on 3 July 2012 [15]. It is the successor to the AMSR-E, which successfully operated for almost ten years from June 2002 to October 2011 [16]. The FY-3B satellite, launched on 5 November 2010, is the second satellite of FY3 (Feng Yun 3) series, a member of China's second generation of polar-orbiting meteorological satellites. It provides measurements of terrestrial, oceanic, and atmospheric parameters, including precipitation rate, sea ice concentration, snow water equivalent, soil moisture, atmospheric cloud water, and water vapor [17]. There was a gap of about ten months between the AMSR-E ceased and the AMSR2 operated, and the Microwave Radiation Imager (MWRI) onboard the FY3B had been running successfully during this period. The AMSR-E, the AMSR2 and the FY3B/MWRI all provide soil moisture products based on X-band. The difference between equatorial local crossing times (the GCOM-W1 at 1:30 a.m./p.m. and the FY3B at 1:38 a.m./p.m.) is within 10 minutes. FY3B/MWRI can fill up the window period if the consistency can be confirmed. Therefore, the evaluation of AMSR2 and FY3B soil moisture products to obtain continuous data on global soil moisture monitoring by the same type of microwave radiometer is of great significance for global water cycle monitoring and long-term continuous monitoring of climate change [18].

In recent years, there have been many studies on the evaluation of soil moisture products. They use a variety of error analysis methods to compare the performance of various products and algorithms based on single site or multiple site data at local or global scales [19–24]. However, due to the complexity of surface soil moisture in the temporal and spatial changes of the surface, experiments at this stage are still insufficient to confirm the superiority and inferiority of various soil moisture products. As the environmental factors change, the performance of soil moisture products may reverse the change results [25].

An in situ soil moisture observation network at pixel scale was constructed in the corn cropland located in the northeast of China, and the experimental period was from early May to late September 2017 which was the only frost-free period in this typical area. In addition, the surface soil structure remained naturally stable without artificial damage during this period. The JAXA and the LPRM soil moisture products from AMSR2 and FY3B/MWRI soil moisture product were evaluated by the performance metrics [26] using the up-scaled in situ soil moisture collected synchronously by the EC-5 probes at 2.5 cm depth. All the three products are all based on X-band where Radio Frequency Interference (RFI) issues are less severe for the X-band soil moisture retrievals than lower frequencies [27,28]. Vegetation has always been one of the main attenuation factors in microwave transfer. Compared to the L-band, the X-band is more susceptible to the surface vegetation cover due to its shorter wavelength [29,30]. The surface temperature also affects the calculation of the vegetation optical thickness and the soil surface emissivity. In the LPRM algorithm, the temperatures of vegetation and soil are approximately equal. In the JAXA algorithm, although the multi-frequency is used to overcome the influence of the surface temperature, the previous research has shown that soil moisture retrieval results were still affected by temperature [25]. Vegetation has always been one of the main attenuation factors in microwave transfer. Compared to the L-band, the X-band is more susceptible to the surface vegetation cover due to its shorter wavelength [29,30]. The microwave

radiation is affected by the temperature. Although the JAXA algorithm hoped to eliminate the effects by frequency difference, the brightness temperature value itself is directly affected by the surface soil temperature (SST). The ranges of products are different, and the field capacity will also vary depending on the climate, environment, and time change. Therefore, the accuracy of soil moisture products under different actual soil moisture conditions may change. The performance of all the products was discussed according to the effect factors including vegetation cover, SST, and actual soil moisture itself in different time periods.

## 2. Materials and Methods

### 2.1. Study Area

As shown Figure 1, the study area is in the north of Changchun City in the northeast of China where the climate is temperate monsoon climate with four distinct seasons. It is a semi-humid area with a flat terrain. The land features are simple, and the main type is cropland with scarce water bodies. The annual sunshine hours are about 2695.2 hours. The average annual precipitation is 520 mm which is mainly concentrated in July and August in summer. The annual average temperature and annual accumulated temperature are 4.4 °C and 2851 °C, respectively. The average daily temperature is below 0 °C from November to March. The temperature difference is large between winter and summer over 50 to 60 °C, and this region is significantly colder than other regions in the same latitude in winter. The frost-free period is about 140 to 150 days from May to September which is also the crop-growing season. The most suitable crop for growing in the region is corn. The study area is a typical and representative cropland in the northeast of China. The research in this paper is of great significance due to the distinctive and specific climate characteristics in the northeast of China unlike the other parts of the world. In this paper, the observation network was established with the SMAP pixel as spatial reference, so that it could include the coverage of pixels of other microwave products. Then the scale of the in situ soil moisture was converted using Thiessen Polygons method to match the pixel size of the target product.

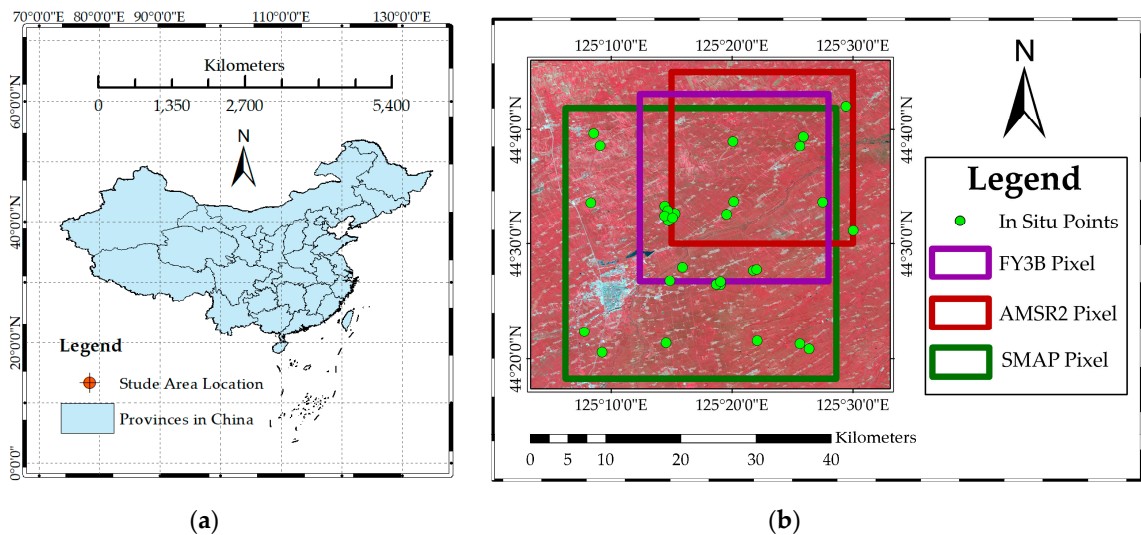

(**a**)                                        (**b**)

**Figure 1.** Study Area. (**a**) shows the geographical location of the study area in the northeast of China; (**b**) shows the distribution of the points from the in situ observation network, the satellite pixels in the study area. The background is a false-color Landsat 8 image at 25 September 2017 with band 5, 4, 3 as the RGB.

### 2.2. Satellite Soil Moisture Products Based on X-Band

Three X-band soil moisture products, the AMSR2/JAXA product, the AMSR2/LPRM product and the FY3B/MWRI product, were selected in this paper. All the three algorithms use a simple

radiative transfer model, the *tau-omega* model [10]. To minimize environmental interference, only the descending products were used while the geophysical conditions were complicated at day times but simple at night times [8,9,31,32].

1.  The AMSR2/JAXA L3 Soil Moisture Product

The AMSR2/JAXA Level 3 0.25° global grid soil moisture product used is acquired at the GCOM-W1 Data Providing Service (https://gcom-w1.jaxa.jp/auth.html). The JAXA algorithm uses a forward radiative transfer scheme to calculate brightness temperatures in multiple frequencies and polarizations according to different vegetation and soil conditions and the surface temperature is assumed constant at 293 K. The soil moisture is estimated by a lookup table built up based on the results and the polarization ratio (PI) at 10.65 GHz and index of soil wetness (ISW) at 36.5 and 10.65 GHz horizontal channels [8,33,34].

2.  The AMSR2/LPRM L3 Soil Moisture Product

The AMSR2/LPRM Level 3 0.25° global grid soil moisture product used is acquired at the Goddard Earth Sciences Data and Information Services Center (GES DISC) (https://gcmd.gsfc.nasa.gov/). The LPRM algorithm is developed by the Vrije Universiteit (VU) University Amsterdam and NASA for multiple frequencies. It uses brightness temperature at 36.5 GHz V channel to estimate land surface temperature and retrieve the soil moisture vegetation optical depth at the same time by an iteration using PI [9,35,36]. For consistency with the other two products, only the X-band LPRM product was used in this paper.

3.  The FY3B/MWRI L2 Soil Moisture Product

The FY3B/MWRI L2 EASE-Grid Soil Moisture Product is acquired at the FENGYUN Satellite Data Center (http://satellite.nsmc.org.cn/). The FY-3B soil moisture retrieval algorithm uses the brightness temperature at 10.65GHz H/V channels based on a parameterized surface emission model (the Qp model) [37] for the bare surface and the empirical relationship between the Normalized Difference Vegetation Index (NDVI) and the vegetation water content to estimate the vegetation optical depth [38] for the vegetation correction.

*2.3. The In Situ Observation Network on Pixel Scale*

2.3.1. Selection of Each Point Location in the In Situ Observation Network

The in situ soil moisture observation network using Decagon EC-5 sensors at pixel scale was constructed to better represent the real surface soil moisture corresponding to the depth of satellite products. At first, the location of each observation site is a key factor in determining whether the observation network is a good representation of the whole experimental area. As shown in Figure 2, a 36 km × 36 km SMAP pixel grid was used as the spatial coverage reference, and the pixel was then subdivided evenly. According to the spatial heterogeneity factors including soil types and bare soil thermal inertia related to bare soil moisture, representative sub-pixels were selected to represent the overall distribution of soil moisture in the whole experimental area, and to minimize the impact of spatial heterogeneity.

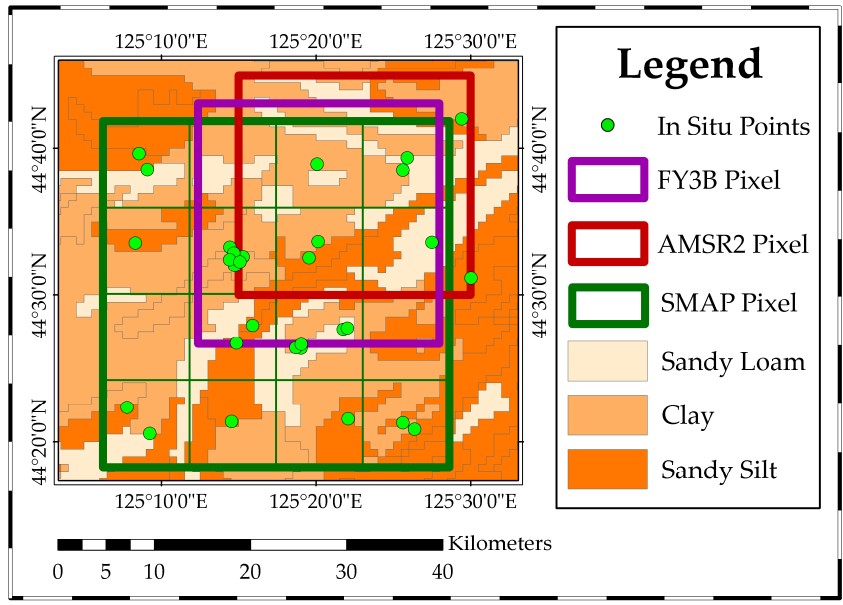

**Figure 2.** Location of each point in the in situ observation network and the distribution of soil types in the experimental area.

### 2.3.2. The Sensor Tests of the In Situ Soil Moisture

The Decagon EC-5 probe was selected due to its effective soil moisture measurement at shallow depths as close as possible to satellite data. The measured soil moisture data were hourly collected from in situ points distributed in the study area. The parameters of all the EC-5 probes had been previously tested and confirmed through:

- The sensing boundary test
- The consistency test
- The calibration according to actual soil from study area

The sensing boundary of the EC-5 probe was confirmed in the laboratory with dry sand and water. The experimental apparatus is shown in Figure 3. A plastic cylindrical container with a height of 30 cm and a diameter of 15 cm is filled with dry sand and placed in a big container filled with water surrounding the plastic container. The EC-5 probe was inserted into the dry sand completely, and gradually collected data from the edge to the center of the container. With a movement interval of 0.5 cm, the data were collected for five times at each position, and the time interval between each data collection was one minute.

The experimental results are shown in Figure 4. During the process of the probe moving from the distance of 0.5 cm to 2.5 cm, the voltage measurement value becomes significantly smaller as the distance becomes larger. Between 2.5 cm and 3 cm, the voltage decreases significantly; after 3 cm, the voltage measurement remains stable. This explains that the probe's boundary measurement range is 2.5–3 cm around.

Due to the precisions of the EC-5 sensors, there existed subtle differences in measurement precision and range between the untested probes. To minimize these differences, we performed a consistency test on all EC-5 sensors with ethanol and dry sand. All the EC-5 soil moisture sensor probes synchronously collected data in ethanol then dry sand at the same temperature. The data sampling rate was once every minute, and the total sampling time was 10 to 15 minutes. The measurement results in ethanol and dry sand are shown in Figure 5a,b. Then the data average of each probe in ethanol and dry sand were calculated. Based on the theoretical values of ethanol and dry sand, the probe with the average value closest to the standard value was selected as the standard probe to correct all the other probes.

The results after correction are shown in Figure 5c,d. The error of the measurement baseline of each probe was significantly reduced after the consistency test.

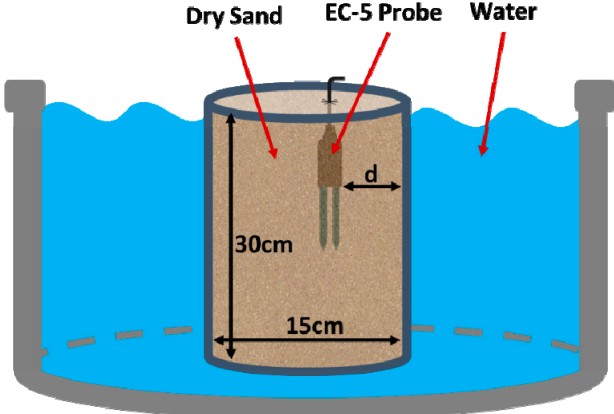

**Figure 3.** Inductive boundary test of EC-5 sensor. The plastic cylindrical container with a height of 30 cm and a diameter of 15 cm was filled with dry sand and placed in a bigger container filled with water to surround the plastic container. Insert the EC-5 probe into the dry sand completely, and gradually collect data from the edge to the center of the container. d was the distance from the probe to the edge of the small container.

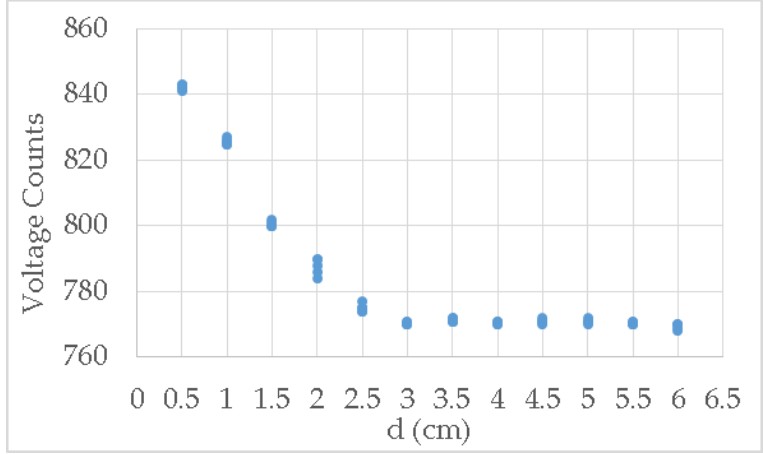

**Figure 4.** EC-5 sensor probe boundary measurement range experiment, the probe measurement voltage value changes with the probe distance d from the dry sand container boundary.

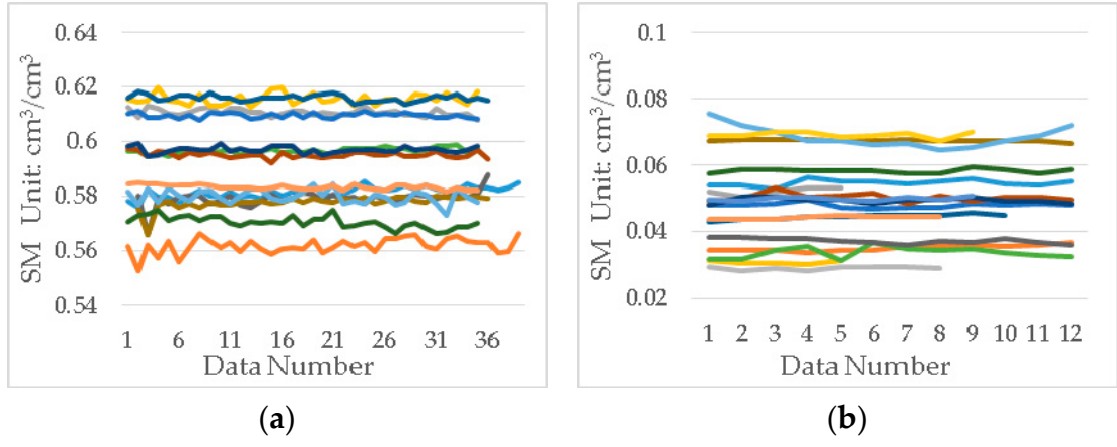

**Figure 5.** *Cont.*

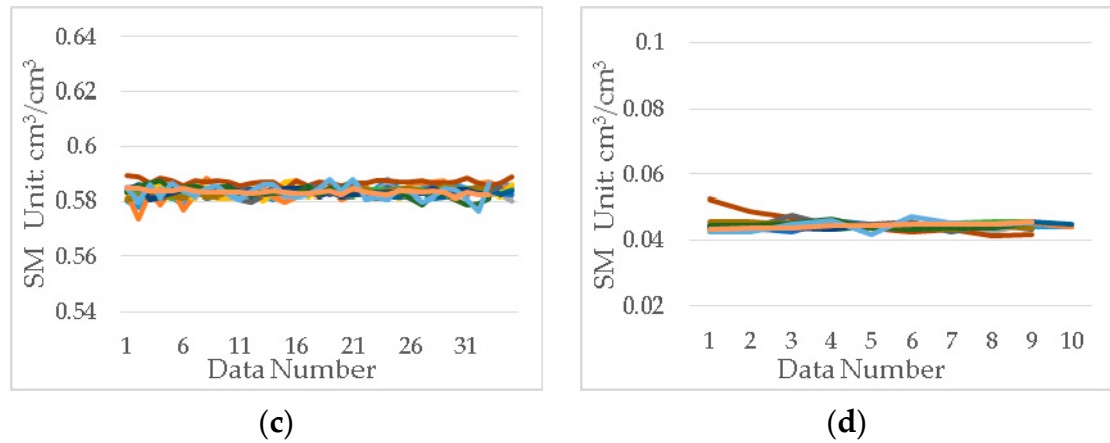

**Figure 5.** Consistency comparison and correction results of EC-5 sensors in ethyl alcohol and dry sand measurements, and different colored lines represent different sensors. (**a**) shows the data collected in ethanol, (**b**) shows the data collected in dry sand. (**c**) shows the ethanol data after correction, and (**d**) shows the dry sand data after correction.

The consistency-tested EC-5 sensors met the uniform accuracy requirements of measurement, but the same sensor probe's response to soil moisture would still vary attributed to different soil types. To acquire accurate in situ data of soil moisture, we calibrated the sensors according to the actual soil samples collected in the study area. Subject to the soil components based on the Harmonized World Soil Database (described in Section 2.4.3), the soil in the study area was divided into sandy loam soil, clay soil, and sandy silt soil, of which the specific contents are shown in Table 1.

**Table 1.** Content of each component in different classification of soil samples.

| Soil Texture | Clay (%) | Silt (%) | Sand (%) |
| --- | --- | --- | --- |
| Sandy Loam Soil | 12.41 | 64.28 | 23.31 |
| Clay Soil | 11.80 | 57.71 | 30.48 |
| Sandy Silt Soil | 11.81 | 55.87 | 32.32 |

Firstly, various types of soil samples collected in the field need pretreatment, and the soil samples were dried at 105 °C for 48 hours, and the dried soil samples were ground and sieved to remove debris such as stones. The sieve pore size is not less than the specific soil type particle size. Then, the pretreated soil was filled into a container (50 cm × 50 cm × 40 cm in volume) and was slowly sprayed with fresh water about 10% of the weight of the soil. The watered soil was kept being stirred to mix evenly. After that, the uniformly mixed wet soil was filled into a small container (13 cm × 13 cm × 15 cm in volume) in a natural state without pressing. The standard EC-5 probe selected in the consistency test was used for calibration. The probe was vertically inserted into the soil at a position greater than 3 cm from the container wall and totally collected 5 data with a sampling interval of 60 seconds. Then a soil sample was taken using a cutting ring (100 cm$^3$ in volume) at its adjacent position and was weighed as its fresh weight. After that, the remaining soil was put back into the container used for mixing. Then all the procedures above were repeated until the soil moisture content was saturated. Finally, all the soil samples collected with cutting ring were dried (105 °C, 48 hours) and weighed. We calculated the volumetric water content of the soil samples, then linearly fit the probe readings corresponding to soil moisture to obtain the calibration parameters and equations of the EC-5 sensor to the three soil types in the study area as shown in Figure 6.

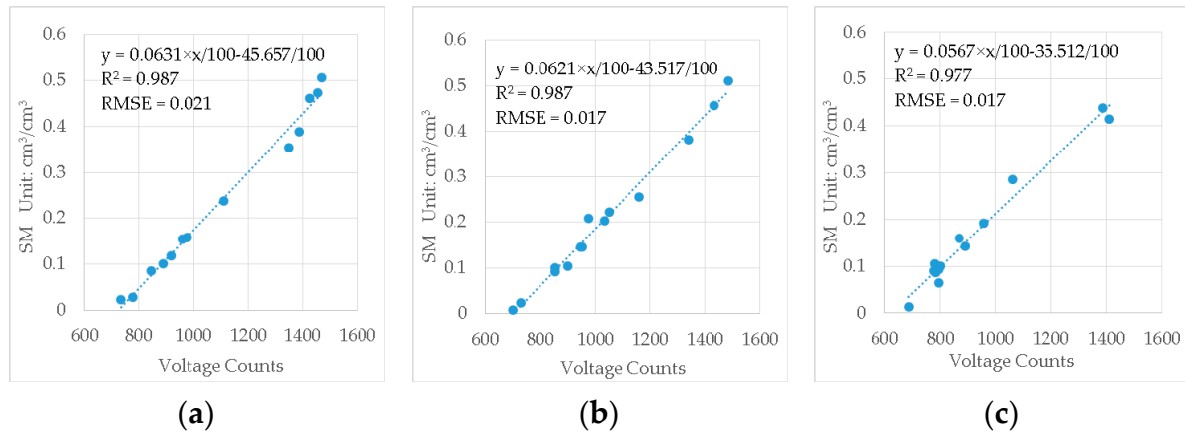

**Figure 6.** The calibration parameters and equations of the EC-5 sensor to the three soil types in the study area, wherein (**a**) is clay soil, (**b**) is sand silt soil, and (**c**) is sandy loam soil.

The accuracies after calibration for different soil types are 0.021 cm$^3$/cm$^3$, 0.017 cm$^3$/cm$^3$ and 0.017 cm$^3$/cm$^3$, which are better than the standard accuracy (0.02 cm$^3$/cm$^3$) of the EC-5 sensor. The specific parameters of the EC-5 sensors after testing and calibration are listed in Table 2.

**Table 2.** Parameters of the EC-5 sensor after testing and calibration.

| Sensing Range (cm) | 2.5~3 |
|---|---|
| Operating Temperature (°C) | −40~+60 |
| Measurement Range of SM (%) | 0~100 |
| Accuracy (cm$^3$/cm$^3$) | 0.02 |

### 2.3.3. The In Situ SST

The temperature sensors used to measure SST were the DS18B20 soil temperature sensor with a range of −55 °C to +125 °C and an accuracy of ± 0.5 °C. Similarly, the DS18B20 temperature sensors were also installed 2.5 cm below soil surface like EC-5 probes. To avoid measurement interference between them and to ensure that the two measurements represent the same position, the DS18B20 temperature sensor should be installed within 5 to 15 cm from the EC-5 probe.

### 2.3.4. Placement of Sensors at In Situ Points

All the sensors were laid in the field at the middle of May after all the land in the experimental area was fully cultivated and were retrieved at the end of September before harvesting. It was also in the frost-free period at this time, which ensured the measured data valid. The surface soil structure remained naturally stable without artificial damage. To exclude the influence of other factors, the sensors were placed under the plain surface of pure soil at 2.5 cm depth that was more than 40 cm away from the plant seed position. As shown in Figure 7a,b, a section was dug next to the selected position and measured with a ruler. The sensor probe was horizontally inserted into the soil at 2.5 cm from the upper soil surface so as not to damage the natural structure of the soil in the vertical direction. The host was buried aside as shown in Figure 7c. The specific terrain of the agricultural land in the study area was the alignment arrangement of ditches and ridges. As shown in Figure 7d, two EC-5 sensors were separately placed at the ditch and the ridge of each in situ point to enable the collected soil moisture data to be more representative. The data were recorded once every hour.

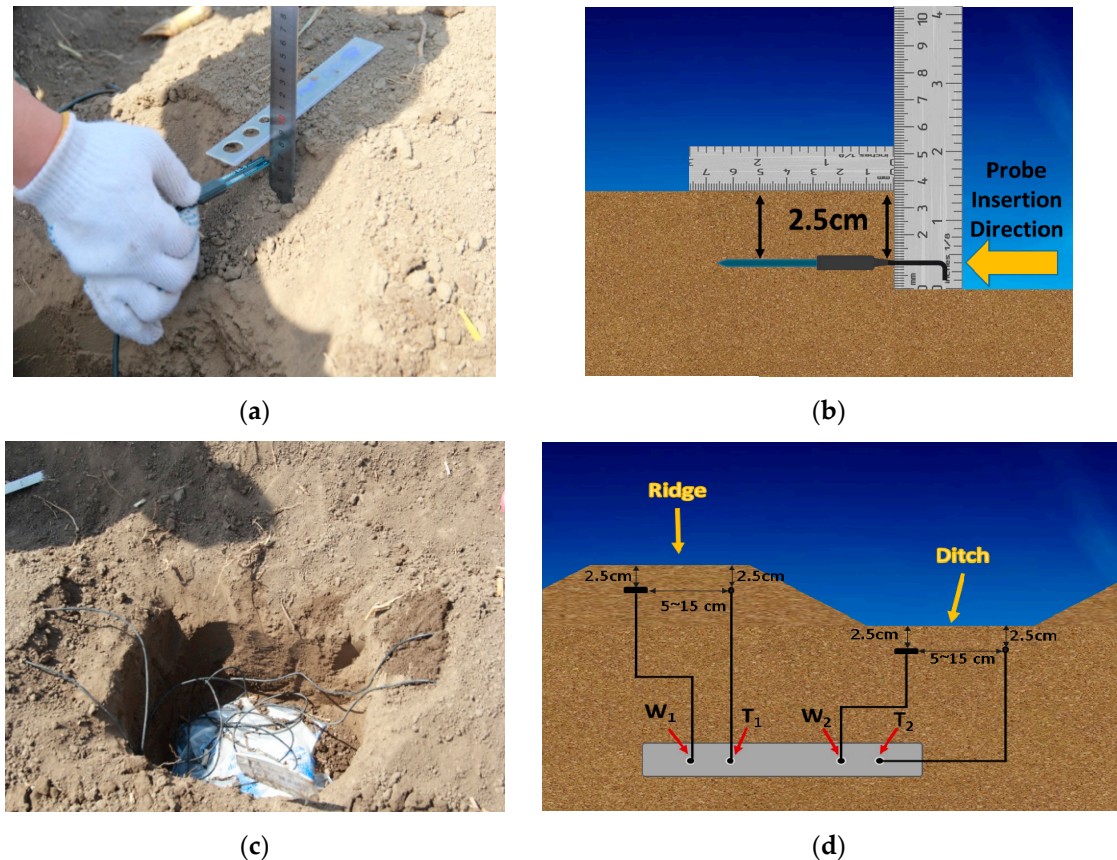

**Figure 7.** Installation and arrangement of the soil moisture and temperature sensors at in situ points. (**a**) shows the actual installation of the EC-5 probe, (**b**) describes the specific installation details of the EC-5 probe, (**c**) shows the situation of host and probes after installation, and (**d**) shows the position details of the probes and host. $W_1$ and $W_2$ are the EC-5 probes, and $T_1$ and $T_2$ are the temperature sensors.

### 2.4. Ancillary Data

#### 2.4.1. Meteorological Data

The global meteorological station's timed observation data were downloaded at the National Meteorological Information Center (http://data.cma.cn/data/cdcdetail/dataCode/A.0013.0001.html), the site data is updated every three hours. With the satellite equatorial crossing time (1:30 A.M.) as the node, all precipitation in the past 24 hours was accumulated as the daily precipitation data.

#### 2.4.2. The Moderate Resolution Imaging Spectroradiometer (MODIS) Vegetation Index

The NASA vegetation index product, MOD13C1 VIs 16-day 0.05deg data, was used to represent the surface vegetation cover situation in the study area [39]. According to different calculation formulas, there are two vegetation indices in this product, the NDVI and the Enhanced Vegetation Index (EVI). The original 0.05° resolution product was resampled to 0.25° by taking the average value. As shown in Figure 8, the trend and amplitude of NDVI and EVI were basically the same, but their change intervals were different, and the correlation between them was significant in the study area.

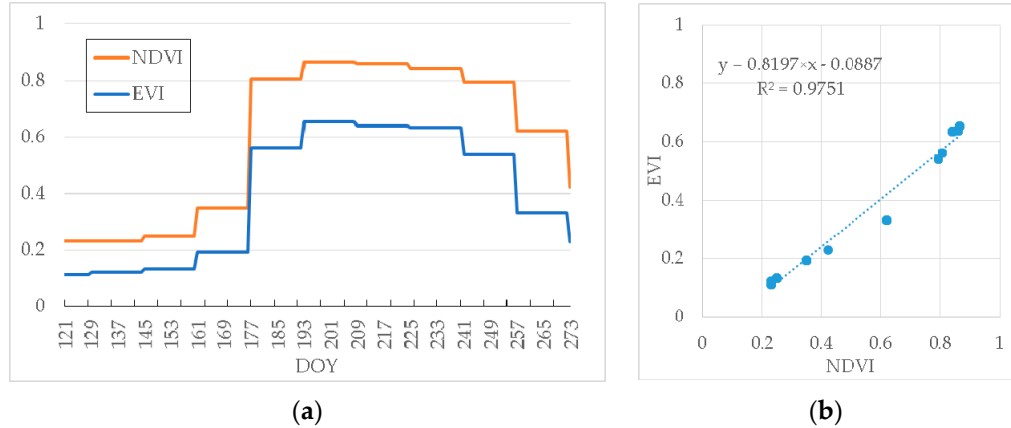

**Figure 8.** The comparison of NDVI and EVI in the study area during the study period. (**a**) shows the changes of NDVI and EVI during the study period, and (**b**) shows a linear relationship between NDVI and EVI.

The EVI was mainly used for analysis in this paper which exhibited the EVI changing obviously. The EVI was at a low level at first, then sharply increased, and maintained at a high level. At the end of the study period, the EVI showed a gradual decline.

### 2.4.3. The Harmonized World Soil Database

The Harmonized World Soil Database is the result of a collaboration between the FAO with IIASA, ISRIC-World Soil Information, Institute of Soil Science, Chinese Academy of Sciences (ISSCAS), and the Joint Research Centre of the European Commission (JRC). It is a 30 arc-second raster database with over 15,000 different soil mapping units [8]. In this paper, we used it to identify the soil types in the study area according to its texture and components.

### *2.5. Methodology*

### 2.5.1. Thiessen Polygons Method for Pixel Scale Matching of the In Situ Data

The pixel size of the passive microwave soil moisture product used in this paper was 25 km × 25 km. To make the point data well represent the actual soil moisture of the whole passive microwave data pixel, the point data, including surface soil moisture and SST, were all up-scaled using Thiessen Polygons (TP) method [40,41] and compared with directly averaged value. As shown in Figure 9, the Thiessen Polygons method divided the entire target pixel area into several polygons according to the position of each point. Every edge of the polygon is in the middle of two points and is perpendicular to the connecting line between the two points. The ratio of the polygon to the total area is the weight of the center point. The sum of all weighted point data is the value of the entire target pixel. The advantages of the Thiessen Polygons method are the simple operation, smooth interpolation results, and basically closed contours generated. Its disadvantage is that it is greatly affected by the known points and only considers the factor of distance. The Thiessen Polygons (TP) approach as an up-scaled method takes the spatial distribution of in situ points into account, while direct average treats the proportion of each point equally. The difference between them is small when the distribution of the points is uniform. However, when some points are gathered together and they have a significant spatial difference from other locations in the pixel, the direct average will be overestimated or underestimated because of the high or low data of the gathered points. However, it could be avoided by the TP approach.

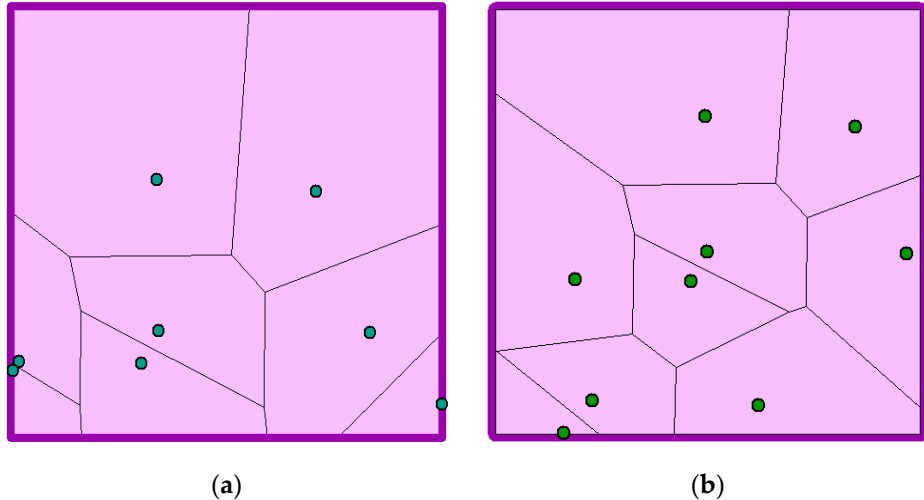

**Figure 9.** Thiessen Polygons calculated from the spatial distribution of the in situ points of soil moisture in microwave pixels, where (**a**) is the AMSR2 pixel with a spatial resolution of 0.25 degrees and a total of 8 in situ points, and (**b**) is the FY3B pixel with a spatial resolution of 0.25 degrees and a total of 9 in situ points.

### 2.5.2. The Performance Metrics for the Evaluation with In Situ Data

The performance metrics including the root mean square error (RMSE), the unbiased root mean square error (ubRMSE), the bias (*b*) and the correlation coefficient (*R*) were used to evaluate the soil moisture products [26]. The formulas are expressed as follows:

$$\mathrm{RMSE} = \sqrt{E\left[\left(SM_{\mathrm{pro}} - SM_{\mathrm{insitu}}\right)^2\right]} \tag{1}$$

$$\mathrm{ubRMSE} = \sqrt{E\left\{\left[\left(SM_{\mathrm{pro}} - E\left[SM_{\mathrm{pro}}\right]\right) - \left(SM_{\mathrm{insitu}} - E\left[SM_{\mathrm{insitu}}\right]\right)\right]^2\right\}} \tag{2}$$

$$b = E\left[SM_{\mathrm{pro}}\right] - E\left[SM_{\mathrm{insitu}}\right] \tag{3}$$

$$R = \frac{\sum\limits_{i=1}^{n}\left(SM_{pro_i} - E\left[SM_{pro}\right]\right)\left(SM_{insitu_i} - E\left[SM_{insitu}\right]\right)}{\sqrt{\sum\limits_{i=1}^{n}\left(SM_{pro_i} - E\left[SM_{pro}\right]\right)^2 \cdot \sum\limits_{i=1}^{n}\left(SM_{insitu_i} - E\left[SM_{insitu}\right]\right)^2}} \tag{4}$$

where $E[\cdot]$ is the expected or linear average operator, $SM_{\mathrm{pro}}$ represents the passive microwave remote sensing soil moisture product estimate, and $SM_{\mathrm{insitu}}$ represents the ascending scale measured soil moisture. *i* means the data number.

## 3. Results

### 3.1. In situ *Soil Moisture Data from the Network*

All the in situ soil moisture data, separately for the AMSR2 and FY3B pixels, from the in situ observation network are shown in Figure 10. The effective period of the data is from a consecutive period of 141 days from 129th to 269th day of the year (DOY). The numbers shown in the figure represent different sensors. Since the AMSR2 pixel and the FY3B pixel mostly overlap, some sensor data were shared by both. The sensors needed replacing the battery and memory card. To ensure the in situ data were continuously obtained, the data may be collected alternately using different numbered sensors at the same in situ point. It can be seen that the soil moisture at each experimental point shows

a highly distinct correlation with precipitation. The soil moisture at each point shows a sharp rise upon the increase and shows a slow downward trend when decreasing.

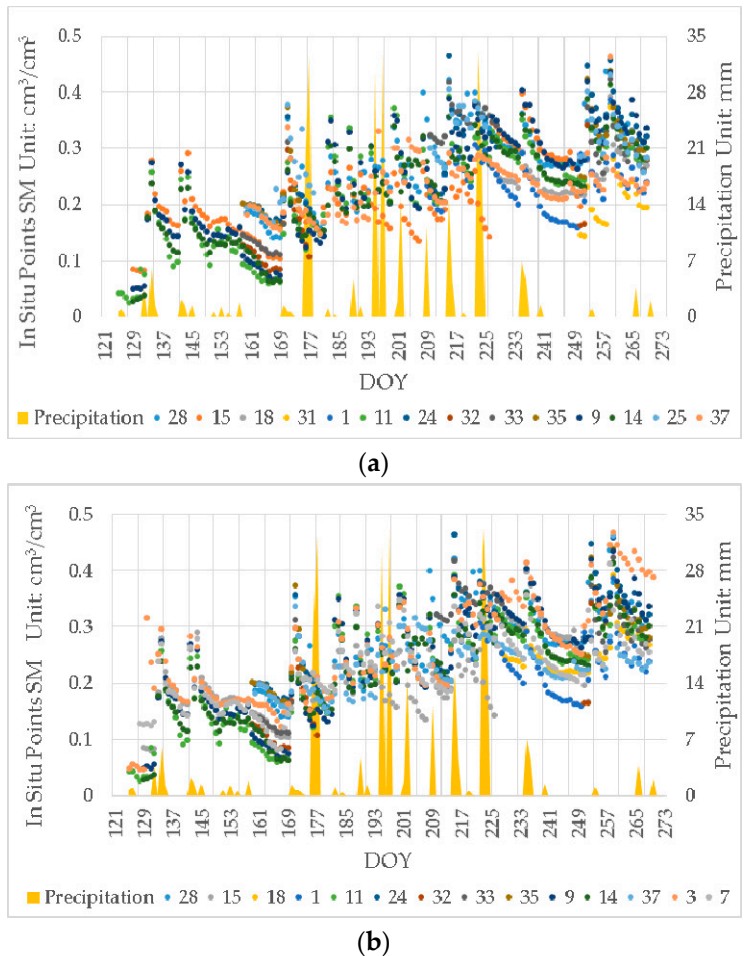

**Figure 10.** Measured soil moisture data from the in situ observation network and the cumulative 24-hour precipitation. Among them, (**a**) is the AMSR2 pixel case, and (**b**) is the FY3B pixel case. The numbers shown at the bottom of the figure are the sensors' numbers.

As shown in Figure 11, the in situ soil moisture data at each point were compared with the up-scaled results. It reveals that the data point from both AMSR2 and FY3B pixel are evenly distributed on both sides of the diagonal, and the direct average is very close to the result of Thiesson Polygons method. That indicates that the in situ points were evenly distributed in the pixel, and the up-scaled average value could well represent the average soil moisture of the entire pixel. There is not much difference between the results of the two methods. The results of Thiessen Polygons method were mainly used as the in situ soil moisture in the later part of this paper. The evaluation results using the direct average will be displayed in Appendix A. Figure 11g shows the comparison of the up-scaled results of AMSR2 pixel and FY3B pixel using Thiessen Polygons method, which reflects that the results are almost the same due to pixels overlapping.

In this experiment, we usually placed more than one sensor in close distance to prevent the individual sensor from failing, and most of the results of the TP approach and the direct average were also very close. To some extent, that indicated that the in situ points were evenly distributed and the spatial variation within the pixel was stable and uniform. However, there were also some large differences between the results of the TP approach and the direct average because of the big changes of the gathered points' data. Because such situations were relatively small, the impacts on the whole time period were limited, so the statistical results also would not be significantly affected. To eliminate

the error caused by the artificial distribution of points, the TP results were preferentially used for calculation. The direct average results were also given as a reference in the attached table.

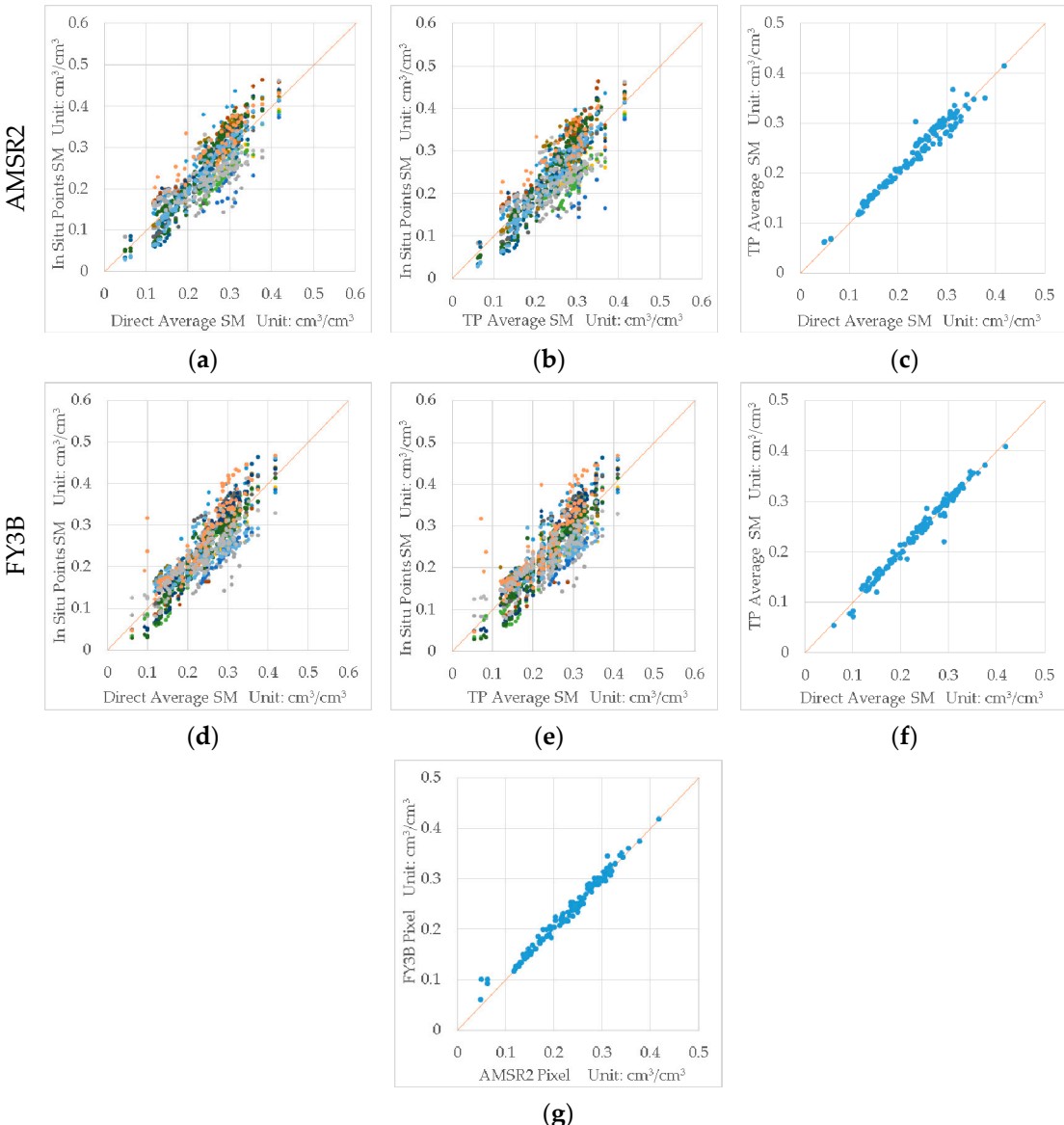

**Figure 11.** The comparison of the in situ point soil moisture data and the up-scaled results in the study area. (**a**,**d**) are respectively the comparison of the in situ soil moisture at each point and the direct average in the AMSR2 pixel and the FY3B pixel. (**b**,**e**) are respectively the comparison of the in situ soil moisture at each point and the result of Thiessen Polygons method in the AMSR2 pixel and the FY3B pixel. (**c**,**f**) are respectively the comparison of the direct average and the result of Thiessen Polygons method in the AMSR2 pixel and the FY3B pixel. (**g**) is the comparison of the results of Thiessen Polygons method in AMSR2 pixel and FY3B pixel.

The experimental period was 141 days, nearly four and a half months, and the seasons spanned spring, summer, and autumn when the vegetation cover and temperature changed greatly. Therefore, according to the objective environmental conditions, mainly seasonal vegetation cover, the whole experimental period was divided into two stages with the DOY of 169 as the node. The first stage was the DOY from 129th to 169th day. At this time, the study area was from spring to early summer, and the surface was mostly from bare soil to low vegetation. The second stage was the DOY from 170th to

269th day. During that period, the season was summer and ended at early autumn while the EVI was high and the land surface was densely covered by vegetation.

As shown in Figure 12, the distribution of the in situ soil moisture of the AMSR2 and FY3B pixels are totally different at different stages. It manifests that the in situ soil moisture is clearly divided into two parts at different stages. Soil moisture in both stages shows a downward trend with increasing temperature. It also can be found that there are two parallel lines separately on the upper side of the data at the first stage and on the bottom side of the data at the second stage, and there is an obvious space between the two sides. The data on the upper side correspond to the maximum value of soil moisture at a specific temperature at the first stage, and the data on the bottom side correspond to the minimum value of soil moisture at a specific temperature at the second stage. The main difference between the two stages was the EVI. At the first stage, the soil can no longer absorb water after soil moisture increased to a certain value because of its yield capacity. Given the land surface of bare soil without vegetation, the yield capacity was controlled by the soil itself and was inversely proportional to the SST. The excessive rainwater would flow away with surface runoff. At the second stage, it was summer with abundant sunshine and high temperature, when much dense vegetation was covering the surface. By virtue of the evapotranspiration, the vegetation roots had a locking effect on soil moisture, which would greatly increase the yield capacity. Therefore, it maintained the lowest value of soil moisture change, and the lowest value was also negatively correlated with the SST. This was mainly resulted from the evapotranspiration of vegetation. The high temperature strengthened the evapotranspiration, causing a decrease in surface soil moisture in the root zone. When the temperature was low, sometimes accompanied by precipitation events, the evapotranspiration became weaker, causing the surface soil moisture to rise.

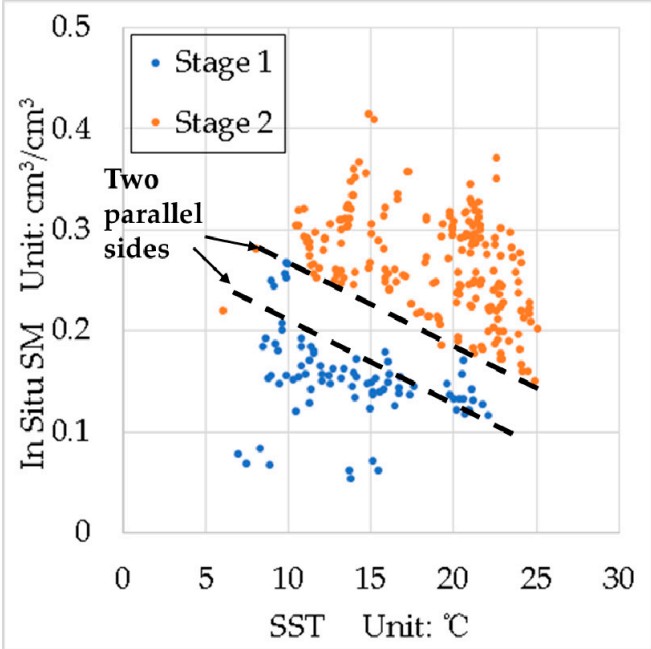

**Figure 12.** The relationship between the in situ soil moisture and the SST. Both of the in situ soil moisture in the AMSR2 and FY3B pixels are shown in the figure at two different stages. There are two parallel lines separately on the upper side of the data at the first stage and on the bottom side of the data at the second stage, and there is an obvious space between the two sides.

### 3.2. Satellite Data Evaluation and Intercomparison

Figure 13 shows the change of the JAXA, LPRM, and FY3B soil moisture products, daily precipitation, and the EVI with the DOY in the study area. It can be seen that the in situ soil moisture is sensitive to precipitation events, with which the local peaks of the soil moisture always appear.

The increasing magnitude of the soil moisture is also consistent with the amount of precipitation at the first stage. The in situ soil moisture generally shows an upward trend during the whole experimental period. The EVI remained at a low level at the first stage while it was spring, and the surface was mostly bare soil. Then the EVI increased slightly from the 161st to the 176th day when the season was late spring and early summer and the surface was covered with low vegetation, but the EVI still remains at a low level. After that, the EVI increased sharply in the DOY from 177th to 192th day when the season was summer, and the vegetation grew densely on the land surface. In the last few periods of the experimental period, the EVI decreased significantly, but it was still significantly higher than that at the first stage. The JAXA soil moisture product and LPRM soil moisture product well cover the experimental period. Among them, JAXA soil moisture products are relatively close to the in situ soil moisture at the first stage and generally less at later. The LPRM product is consistently higher than the in situ soil moisture all the period and the changes are severe. The FY3B soil moisture product are severely deficient in the experimental period, and the data are cut off from the 161th of DOY and recover from the 256th of DOY. The large amount of missing data of the FY3B product is mainly concentrated in the period with high EVI, high SST, and high precipitation in summer. Moreover, the brightness temperature data of FY3B at 10 GHz are intact and very close to the AMSR2 data. Therefore, it can be judged that the missing of the FY3B product is mainly due to the FY3B soil moisture product algorithm. The maximum of the FY3B product is 0.5 cm$^3$/cm$^3$. In previous studies, it was common for the algorithms to be saturated or even overflowed under dense vegetation cover. [42,43].

The precipitation data from the meteorological station is the cumulative amount every 3 hours. The daily precipitation data in this paper was the accumulated precipitation data from the past 24 hours since the satellite transited. The large amount of precipitation that occurred in 176 days was mainly concentrated at 3 to 6 o'clock, which was a short-term concentrated precipitation. The satellite transit time was around 17:30, with a gap of 9.5–12.5 hours. The in situ data showed that most of the in situ points barely reacted. The meteorological station is about 60km away from the study area, and the study area covered 36 km×36 km. It may be that the precipitation in the area was not obvious. On the other hand, the EVI rose sharply and the crop grew rapidly at this time. The surface temperature was also at the highest value during the experimental period. So, the evaporation and transpiration cannot be ignored. The moisture might be reduced largely before the satellite transited. Therefore, for all the analysis above, the high precipitation at 176[th] day did not lead to an increase in soil moisture.

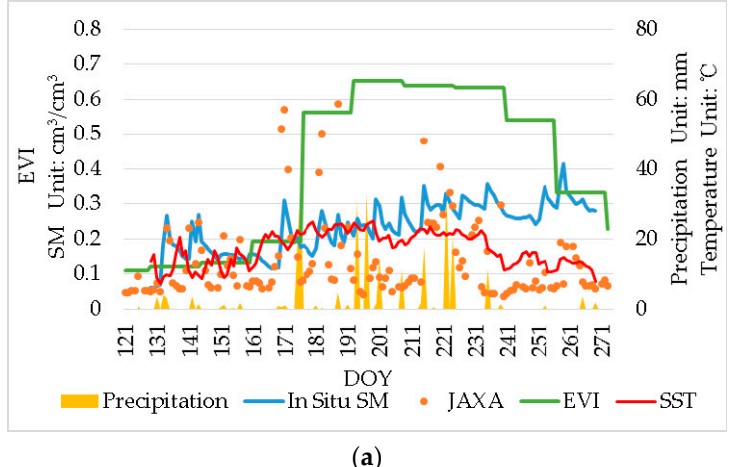

(**a**)

**Figure 13.** *Cont*.

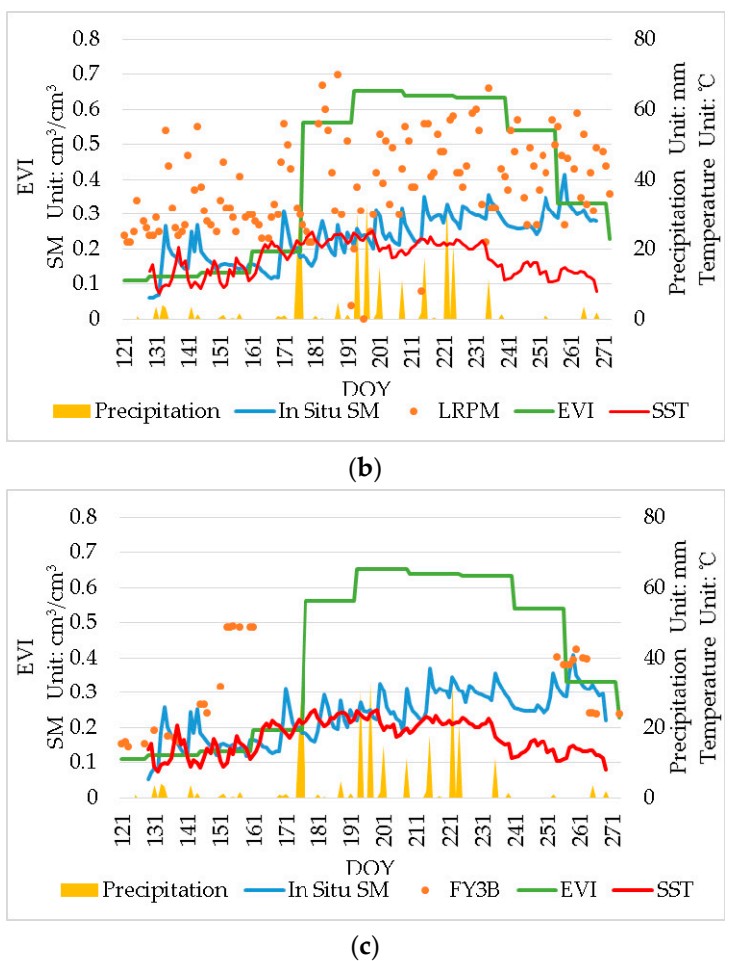

**Figure 13.** The change of soil moisture products with in situ soil moisture, daily precipitation, EVI, and surface soil temperature. Among them, (**a**) is about the JAXA soil moisture product, (**b**) is about the LPRM soil moisture product, and (**c**) is about the FY3B soil moisture product.

Figure 14 shows the comparisons among the JAXA product, the LPRM product and the in situ soil moisture in different periods, and Table 3 lists the results of the performance metrics including the RMSE, the ubRMSE, the *b*, and the *R*. The results using the direct average is displayed in Table A1 in Appendix A. It exhibits that the JAXA product generally shows an underestimation of the soil moisture with a *b* of $-0.094$ cm$^3$/cm$^3$. In contrast, the LPRM product generally demonstrates a large overestimation of soil moisture with a bias of 0.156 cm$^3$/cm$^3$. The JAXA's RMSE is 0.150 cm$^3$/cm$^3$ that is smaller than the LPRM's 0.191 cm$^3$/cm$^3$ but the ubRMSE of them are almost the same during the whole period. Both the JAXA product and the LPRM product are close to the in situ data at the first stage. The JAXA points are evenly distributed on both sides of the diagonal while the in situ soil moisture is low in this period. However, the LRPM product has already provided an obvious overestimation of soil moisture at this time. At the second stage, the JAXA product displays a significantly underestimation when the in situ soil moisture overall increases, and there are also some overestimated data. All the performance metrics of the two products become worse except for the LPRM's *b*. However, the LPRM still keep an overestimation of the soil moisture at a high level. The JAXA product has the best performance of error with the lowest RMSE and ubRMSE of 0.066 cm$^3$/cm$^3$ and 0.049 cm$^3$/cm$^3$, and the *b* is $-0.043$ cm$^3$/cm$^3$ that reaches the goal accuracy of the product ($\pm 0.05$ cm$^3$/cm$^3$). However, the LPRM product has the highest correlation coefficient of 0.654 at this time. The JAXA product range is 0 to 0.6 cm$^3$/cm$^3$, while the LPRM product is 0 to 1 cm$^3$/cm$^3$. It can be seen from the linear prediction of the data in Figure 14 that in the two periods, the LPRM product generally shows an upward trend with the increase of the in situ soil moisture.

The  slope of the linear prediction is basically consistent with the range ratio of the LPRM product to the in situ soil moisture. Moreover, the LPRM product has better correlation with in situ soil moisture than the JAXA product throughout the whole experimental period. Comparing the two products, the LPRM product are generally higher than the JAXA's. They show a good linear relationship with the correlation coefficient of 0.94 at the first stage, and the change of the difference between them becomes very severe at the second stage.

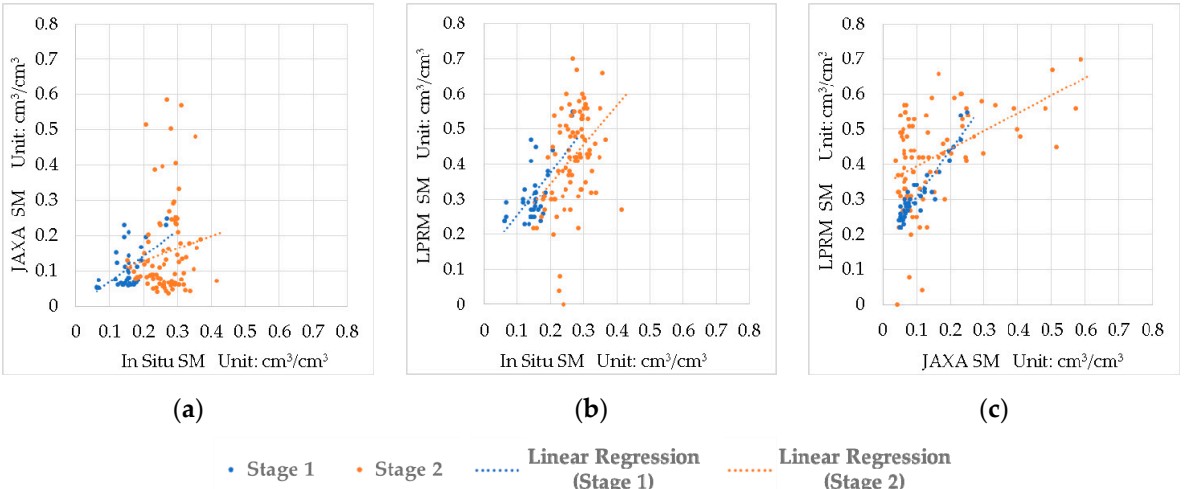

**Figure 14.** Comparisons between the JAXA product, the LPRM product, and the in situ soil moisture in two stages. (**a**) is the comparison between the JAXA product and the in situ soil moisture. (**b**) is the comparison between the LPRM product and the in situ soil moisture. (**c**) is the comparison between the two AMSR2 products.

**Table 3.** The performance metrics of the JAXA and the LPRM soil moisture products at different study periods. The best one for each performance metric is in bold.

| Period | Products | RMSE (cm³/cm³) | ubRMSE (cm³/cm³) | $b$ (cm³/cm³) | $R$ |
|---|---|---|---|---|---|
| Whole Period | JAXA | 0.150 | 0.117 | −0.094 | 0.259 |
| | LPRM | 0.191 | 0.110 | 0.156 | 0.542 |
| First Stage | JAXA | **0.066** | **0.049** | **−0.043** | 0.565 |
| | LPRM | 0.177 | 0.063 | 0.166 | **0.654** |
| Second Stage | JAXA | 0.173 | 0.129 | −0.115 | 0.136 |
| | LPRM | 0.196 | 0.124 | 0.152 | 0.403 |

Due to the serious lack, the FY3B soil moisture product cannot be evaluated throughout the experimental period like the JAXA and the LPRM products. Therefore, the three soil moisture products were evaluated based on the coverage of the FY3B soil moisture product. The results are shown in Table 4. The results using the direct average is displayed in Table A2 in Appendix A. It can be seen that the RMSEs of the three products are almost the same. The results of the FY3B product is the neither the best nor the worst except for the *R*. The LPRM still has the best *R* of 0.516 with a *P*-value of 0.0117 in the significance level. However, due to only 23 days of data in total, the FY3B product does not have a good performance during the study period.

**Table 4.** The comparison of the performance metrics of the three soil moisture products as the period of the FY3B product. The best one for each performance metric is in bold.

| Period | Products | RMSE (cm³/cm³) | ubRMSE (cm³/cm³) | $b$ (cm³/cm³) | $R$ |
|---|---|---|---|---|---|
| **as FY3B** | **FY3B** | 0.237 | 0.155 | 0.179 | 0.042 |
| | **JAXA** | **0.231** | **0.085** | 0.215 | 0.180 |
| | **LPRM** | 0.233 | 0.156 | **0.174** | **0.516** |

## 4. Discussion

The winter in the study area is long and cold, and the frost-free period of the year is only about 5 months. Only in this period, the soil moisture is guaranteed to be liquid and free of ice. Therefore, the local climatic conditions determine the effective monitoring period of soil moisture in this study. The land types and soil types in the study area are stable, most of which are agricultural land. The crops are basically corn. Although the crops in this area only can be planted once a year, the black soil here is fertile and the quality of crops is almost the best in the country. The annual climate change in the study area is basically stable, and the experimental period is representative. Therefore, it is of great significance to monitor the climatic conditions in this region and the soil moisture change during the growing season. The effect of the environment on high-quality crop production can be analyzed by studying the effects of various ecological climate changes in the study area on crop growth. The preceding results reveal obvious differences among the three soil moisture products, and the products also have highly varying performance at different periods. Since the pixels in the study area were monotonous, we analyzed the factors including vegetation, SST, and even the actual soil moisture itself that may affect the products.

*4.1. The Vegetation Cover Effect*

As shown in Figure 15, the EVI, representing the vegetation change is used to be compared with the differences among the products and the in situ soil moisture. With the EVI increasing, only the JAXA product underestimates more obviously. The differences grow between the product and the in situ soil moisture. The differences between the LPRM product and the in situ data almost remain as the EVI rises, but some soil moisture data are clearly underestimated. In the case of lower EVI, the differences between the LPRM product and the JAXA product are concentrated at a lower level. With the increase of the EVI, the vegetation coverage becoming dense, the range of the differences is expanded. According to Figure 13, the EVI keeps rising until declining to 0.33 at the end of the experimental period. Compared with the high EVI (0.54~0.65) in the dense vegetation cover in summer, the EVI has decreased significantly, which is much closer to the spring EVI (0.11~0.19). It manifests that the differences between the products and the in situ data do not grow smaller as the EVI drops to 0.33. In particular, the JAXA product is still similar to the situation with the high EVI in summer. The differences between the three products and the in situ data were analyzed after the recovery of the FY3B product in the later experimental period. It was found that the performance of FY3B seems to be better than the JAXA and LPRM products at this time. The differences between the FY3B product and the in situ soil moisture are completely lower than the JAXA's and basically lower than the LPRM's. In addition, the P values in the significance level were 0.0341 and 0.0001 respectively.

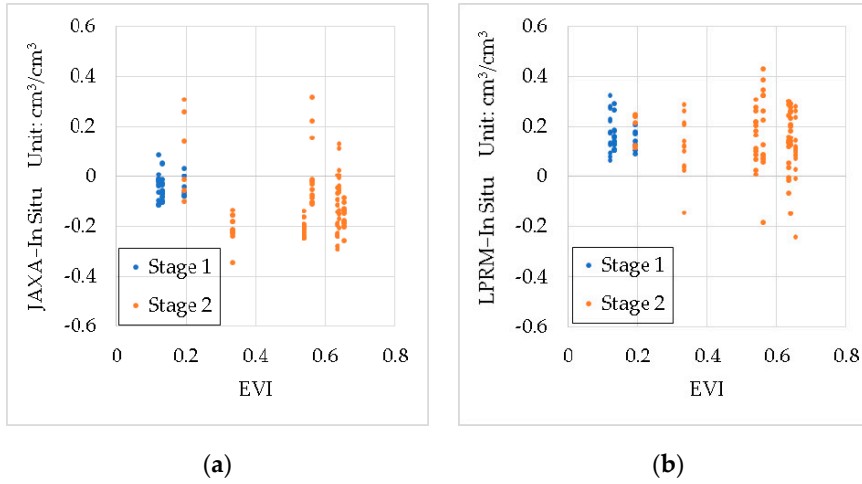

(**a**) (**b**)

**Figure 15.** *Cont.*

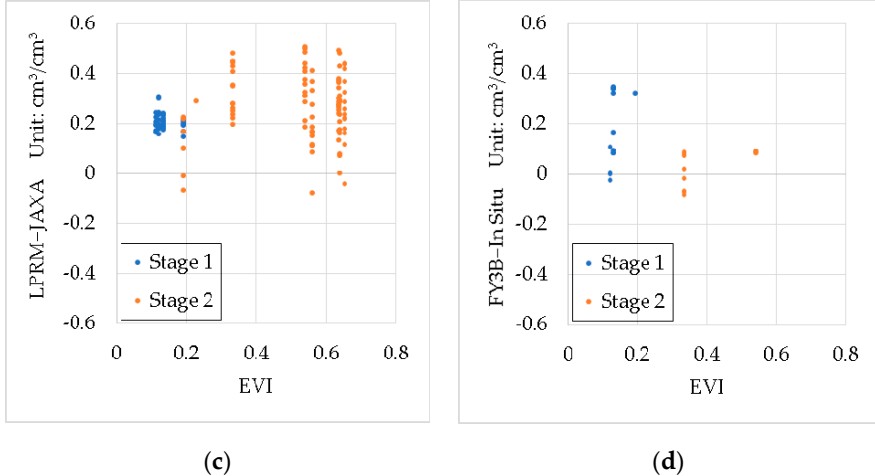

(**c**)  (**d**)

**Figure 15.** The effect of the EVI to the soil moisture products. (**a**) is the difference between the JAXA product and the in situ soil moisture with the EVI, (**b**) is the difference between the LPRM product and the in situ soil moisture with the EVI, (**c**) is the difference between LPRM product and the JAXA product with the EVI, and (**d**) is the difference between the FY3B product and the in situ soil moisture with the EVI.

## 4.2. The Effect of the SST

The SST was in situ measurement like soil moisture, and both were consistent in time and space, which means that the SST is more accurate and reliable. On the other hand, the SST can directly affect the brightness temperature and emissivity of the surface soil, which will affect the retrieval results of soil moisture. Therefore, it is meaningful to study the effects of the SST on the results. The performance of the soil moisture products with the SST is shown in Figure 16. It can be seen that the JAXA product is very close to the in situ soil moisture at the first stage, and its distribution with SST is also similar to the in situ soil moisture's. At the second stage, the JAXA's distribution becomes dispersed with the SST increasing. It is different from the in situ soil moisture in Figure 12, and no obvious bevel edge appears on the lower side. At the same time, the underestimation of the JAXA product with the SST is not stable, and some data even overestimate the soil moisture. At the second stage, the JAXA product has larger gap to the in situ data overall. From the linear prediction in Figure 16a, the JAXA increases with the SST increasing. This offsets the previous underestimation to some extent. Although some of this is caused by precipitation, the LPRM product that are more sensitive to precipitation events has not shown an upward trend at this stage. Therefore, we believe that the soil moisture product of JAXA algorithm is still obviously affected by the SST at this stage. The variation trend of the LPRM product is relatively consistent during the whole experimental period. At the second stage, its distribution also has a fuzzy edge similar to Figure 12. However, the range of the LPRM product is different due to its overall overestimation. In addition, the LPRM product also shows some underestimation as the SST rises. It all occurs at the SST above 15 °C except for one time. The difference between the in situ data and the products all become more dispersed with the increasing SST at the second stage. The difference between the JAXA and the LPRM soil moisture products is stable at the first stage, but it becomes smaller as the SST increases at the second stage. This may be that the JAXA product range is 0 to 0.6 $cm^3/cm^3$, while the LPRM product is 0 to 1 $cm^3/cm^3$. When the actual soil moisture rises, the difference between the two products is enlarged. When there is no rain event accompanied by an increasing SST, both products become smaller, so that the gap between them is shrunk.

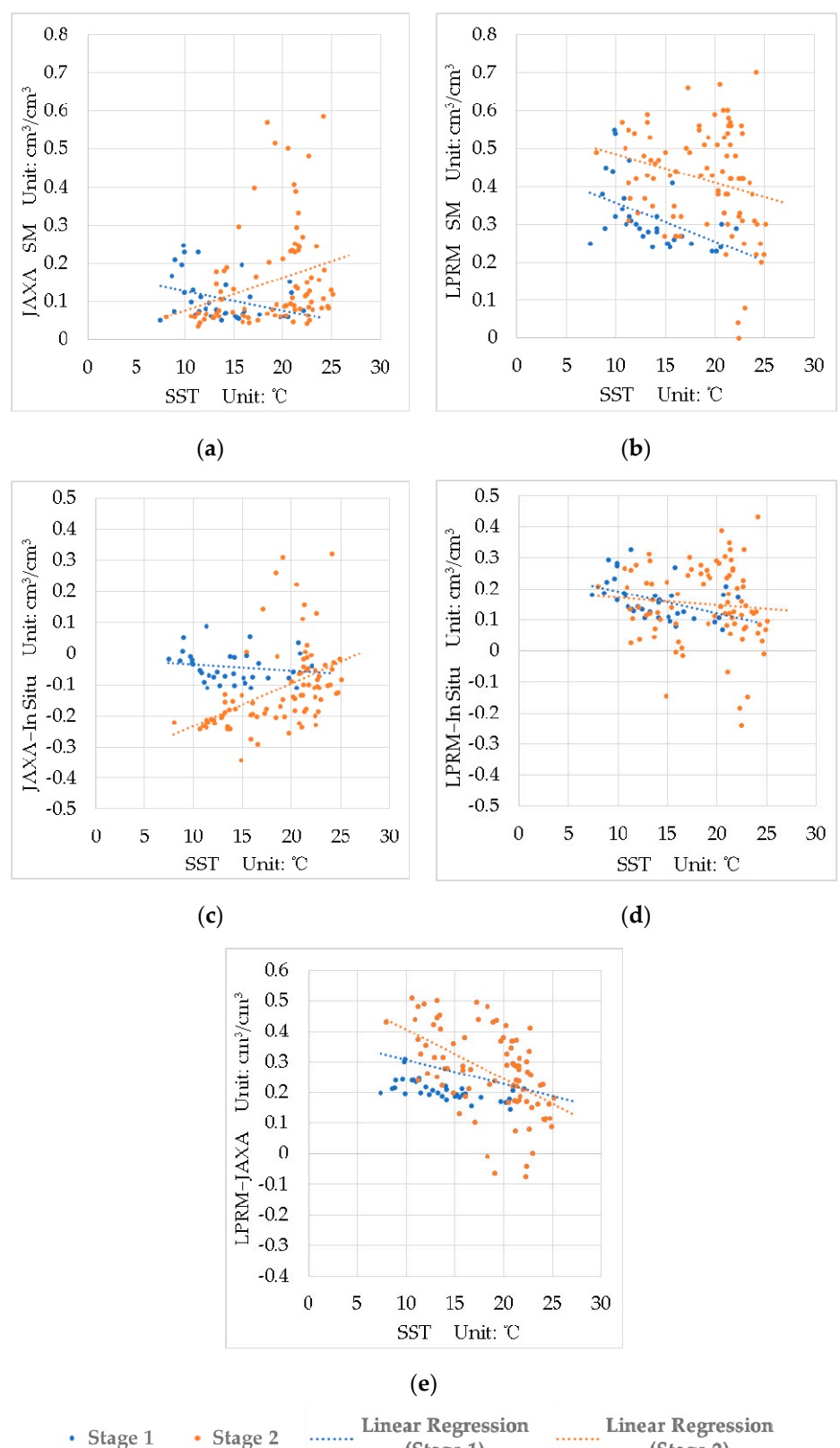

(**a**)  (**b**)

(**c**)  (**d**)

(**e**)

• Stage 1  • Stage 2  ······· Linear Regression (Stage 1)  ······· Linear Regression (Stage 2)

**Figure 16.** The effect of the SST to the soil moisture products. (**a**) shows the JAXA product with the SST, (**b**) shows the LPRM product with the SST, (**c**) shows the difference between the JAXA product and the in situ soil moisture with the SST, (**d**) shows the difference between the LPRM product and the in situ soil moisture with the SST, and (**e**) shows the difference between the LPRM and the JAXA products with the SST.

### 4.3. The Actual Soil Moisture Change

Both the EVI and the SST have a very significant increase from the first stage to the second stage. In addition, the performance of the JAXA product and the LPRM product at the first stage is generally better than that at the second stage. As can be seen from Figure 13, the EVI and the SST decline at the end of the experimental period, but the errors in the JAXA and the LPRM products are not improved apparently. On the other hand, although the in situ soil moisture varies locally, it has been increasing in the overall experimental period.

As shown in Figure 17, the change of the products' errors and the different between the two AMSR2 products are compared with the in situ soil moisture. It can show the impact of the actual surface soil moisture itself on the products' performance more clearly than Figure 14. It can be seen that although some data may be close or overestimated, most of the JAXA product more underestimate the soil moisture as the in situ soil moisture increases. There is a significant bevel edge on the bottom of the distribution of the difference between the JAXA product and the in situ data, which is related to the fact that the product value has been maintained at a low level and the product range is 0 to 0.6 $cm^3/cm^3$. However, the points of scatterplots of the LPRM product does not have such a clear edge. Its range of the variability become larger, and so is its error. However, the range of the in situ soil data is basically below 0.4 $cm^3/cm^3$ that is not beyond the product range. As shown in Figure 17b, the linear predictions of the differences between the LPRM product and the in situ data slightly increase with the actual soil moisture increasing during the both periods. In addition, two slopes in two periods are almost the same. One reason is that the difference between the range of the LPRM product and the field capacity becomes obvious as the actual soil moisture increasing. Previous studies have shown that the LPRM algorithm is very sensitive to the temporal variability of soil moisture, but its absolute accuracy is difficult to guarantee [31,32,44,45]. The LPRM product generally overestimates the soil moisture and is also higher than the JAXA product [24,25]. The reason may be that the soil moisture range of the LPRM algorithm is 0-1 $cm^3/cm^3$, but the field capacity is generally ~0.5 $cm^3/cm^3$ [25], and the LPRM is very sensitive to the temporal variability, so it is more likely to exceed the actual soil moisture. The LPRM product generally has a good correlation with in situ data without considering the absolute accuracy [46]. This may also explain that the overestimation bias of the LPRM product is large, but the correlation is good in this paper. The changes in the two AMSR2 products also lead to an increase and complexity in the difference between the two soil moisture products as the in situ soil moisture increases. After that, even though the EVI and the SST both fall to the level close to the first stage, the performance of the JAXA and the LPRM products are not enhanced. It is worth noting that as shown in Figure 17d, the FY3B product performs better than the JAXA and the LPRM products in the case of high soil moisture. Although the amount of the data is limited, the FY3B product seems to have certain advantages at the end of the experimental range where both the EVI and the SST decrease but the in situ soil moisture continues to rise.

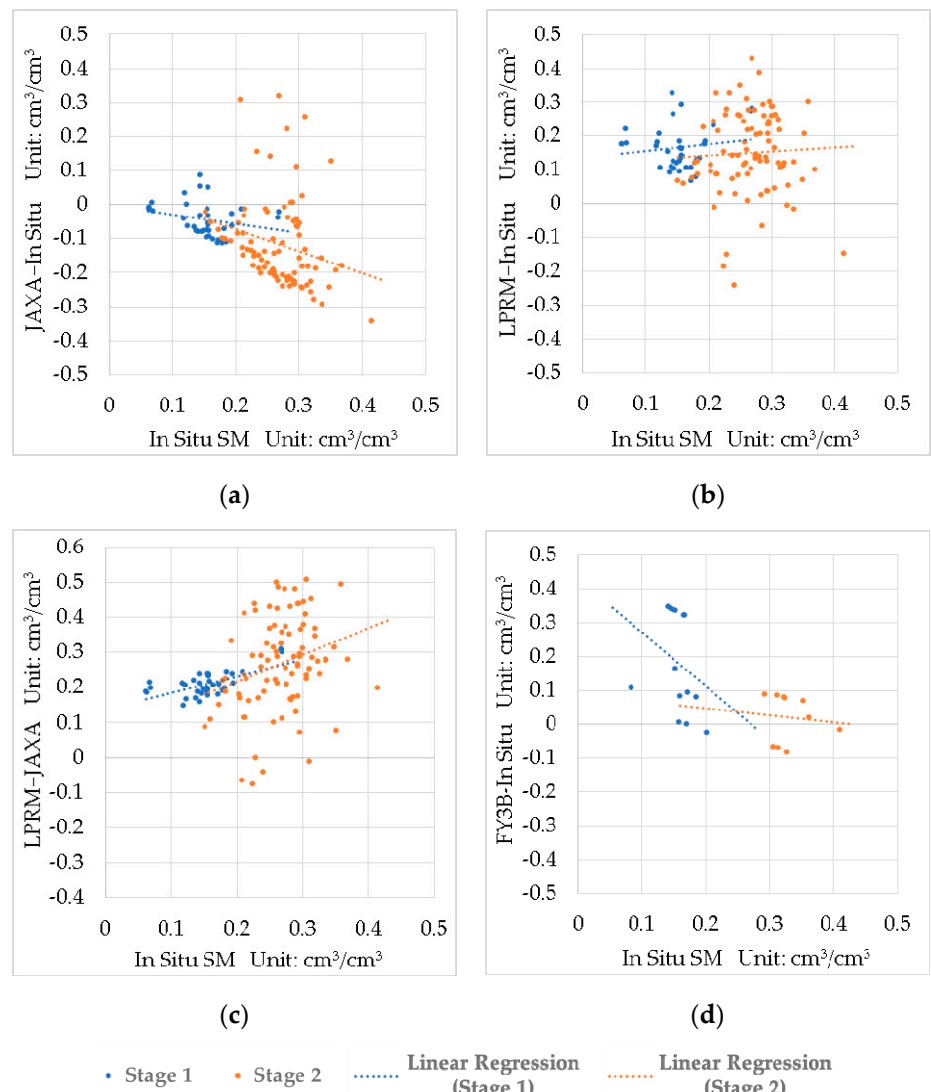

**Figure 17.** The effect of the actual soil moisture to the soil moisture products. (**a**) shows the difference between the JAXA product and the in situ soil moisture with the in situ soil moisture, (**b**) shows the difference between the LPRM product and the in situ soil moisture with the in situ soil moisture, (**c**) shows the difference between the LPRM product and the JAXA product with the in situ soil moisture, and (**d**) is the difference between the FY3B product and the in situ soil moisture with the in situ soil moisture.

## 5. Conclusions

In this paper, an in situ soil moisture observation network in cropland on pixel scale in the northeast of China was designed considering the unique climatic characteristics of the regional area and the detection depth of the satellite sensors. The crop-growing season from May to September 2017, almost covering the whole frost-free period, was selected as the experimental period. Multiple EC-5 soil moisture sensors were arranged in a typical cropland in the northeast of China as the study area to obtain data every hour. All the sensors were calibrated according to the soil in the experimental area, so that the in situ soil moisture was consistent with the satellite products in terms of time, space, and depth. The results showed that JAXA product underestimated with a $b$ of -0.094 cm$^3$/cm$^3$ and the LPRM product seriously overestimated the soil moisture with a $b$ of 0.156 cm$^3$/cm$^3$ throughout the whole experimental period. The FY3B product was severely deficient in the experimental period and was all absent when the EVI was above 0.5. When it was bare soil or less vegetation cover, the JAXA product had the best performance of errors with the lowest ubRMSE at 0.049 cm$^3$/cm$^3$ and the $b$ at

-0.043 cm$^3$/cm$^3$ that reached the goal accuracy of the product ($\pm$0.05 cm$^3$/cm$^3$), and the LPRM product had the best correlation ($R$ = 0.654). When the EVI increased over time, all of them declined. When the EVI was low, the JAXA product errors were less affected by the SST, but the LPRM product slightly decreased with the SST increasing. When the EVI was high, the distribution of the JAXA product errors became more complicated, and some data decreased with the SST increasing, but the LPRM product did not change significantly at this time. Throughout the experimental period, the ranges and errors of the JAXA and LPRM products showed an upward trend, and only the in situ soil moisture showed a similar trend throughout the whole period. Therefore, the JAXA product error was smaller and the LPRM product correlation was better. At the end, the EVI and SST turned to decrease and the in situ soil moisture kept increasing. Although the amount of data was limited, the FY3B product seemed to have a better error performance than the JAXA and LPRM products.

**Author Contributions:** Conceptualization, H.F.; methodology, H.F. and T.Z.; software, H.F. and T.Z.; validation, H.F. and T.Z.; formal analysis, H.F.; investigation, H.F.; resources, C.S.; data curation, H.F.; writing—original draft preparation, H.F.; writing—review and editing, H.F. T.Z. and C.S.; visualization, H.F. and T.Z.; supervision, C.S.; project administration, C.S.; funding acquisition, C.S.

**Funding:** This work was supported by National Natural Science Foundation of China (NSFC) (11574113, 11374123, 11104106); Science and Technology Planning Project of Jilin Province (20180101238JC, 20170204076GX, 20180101006JC, 20190103041JH), China Postdoctoral Science Foundation (BX20180127).

**Acknowledgments:** The first author would like to thank Changchun Jingyuetan Remote Sensing Experiment Station, Chinese Academy of Sciences for the support for the in situ soil moisture and temperature data. The first author especially thank Prof. Xingming Zheng at the Northeast Institute of Geography and Agroecology, Chinese Academy of Sciences. Without his great contribution to the conceptualization, this article cannot be done. The first author also thank Prof. Lingjia Gu at College of Electronic Science and Engineering, Jilin University for her the guidance to this article. The first author also is grateful to Dr. Tao Jiang and Dr. Yu Bai at the Northeast Institute of Geography and Agroecology, Chinese Academy of Sciences for the cooperation in the construction of the In Situ Observation Network.

**Conflicts of Interest:** The authors declare no conflict of interest.

## Appendix A

**Table A1.** The performance metrics of the JAXA and the LPRM soil moisture products at different study periods with the in situ data using direct average. The best one for each performance metric is in bold.

| Period | Products | RMSE (cm$^3$/cm$^3$) | ubRMSE (cm$^3$/cm$^3$) | $b$ (cm$^3$/cm$^3$) | $R$ |
|---|---|---|---|---|---|
| Whole Period | JAXA | 0.144 | 0.113 | −0.090 | 0.321 |
| | LPRM | 0.194 | 0.109 | 0.160 | 0.551 |
| First Stage | JAXA | **0.061** | **0.049** | **−0.037** | 0.413 |
| | LPRM | 0.184 | 0.063 | 0.173 | **0.579** |
| Second Stage | JAXA | 0.167 | 0.124 | −0.112 | 0.252 |
| | LPRM | 0.198 | 0.122 | 0.155 | 0.424 |

**Table A2.** The comparison of the performance metrics of the three soil moisture products as the period of the FY3B product with the in situ data using direct average. The best one for each performance metric is in bold.

| Period | Products | RMSE (cm$^3$/cm$^3$) | ubRMSE (cm$^3$/cm$^3$) | $b$ (cm$^3$/cm$^3$) | $R$ |
|---|---|---|---|---|---|
| **as FY3B** | **FY3B** | 0.236 | 0.155 | 0.178 | 0.018 |
| | **JAXA** | **0.228** | **0.087** | 0.211 | 0.144 |
| | **LPRM** | 0.232 | 0.158 | **0.170** | **0.500** |

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
