# Peer review of "Evaluation and Analysis of AMSR2 and FY3B Soil Moisture Products by an In Situ Network in Cropland on Pixel Scale in the Northeast of China"

_remotesensing, doi:10.3390/rs11070868_

Round 1

Reviewer 1 Report

@page { margin: 0.79in } p { margin-bottom: 0.1in; line-height: 115% }

An in situ soil moisture (SM) network has been set up in corn cropland in northeast China. This network is used for the evaluation of SM datasets derived from AMSR2 (both the LPRM and JAXA products) and FY3B/MWRI. The performance of the three satellite SM datasets is discussed with regards to the vegetation state, surface soil temperature (SST) and in situ SM.

The manuscript is written in an understandable way; some parts require however language checking.

The authors made an effort in installing and calibrating the in situ SM network, and they describe all related steps in detail. However, the evaluation of the satellite SM lacks some methodological aspects and interpretation/discussion. My comments and questions to the authors are as follows:

1) Abstract: you describe what you did and what the resulting metrics are, but you should include at least 1-2 sentences of interpretation of those results. Sentences like “The FY3B product is seriously missing” are confusing, if the reader is not familiar with the data set and the rest of the manuscript.

2) Introduction – last paragraph: formulation of distinct science questions would help (i.e., why was the performance of the products discussed according to vegetation, SST and actual SM).

3) Fig. 1b seems to be wrong – the caption refers to Fig. 2. An idea would be to keep Fig. 1b but instead of the relatively homogeneous (and therefore not very interesting) Landsat image display land cover in the background. Why is the SMAP pixel shown in Figs. 1 and 2?

4) Fig. 4, but relevant throughout the manuscript: Using the same layout and design style for all figures would make the manuscript look more professional. Right now, it looks as if a number of people have worked on the paper who just put their results together without any final layout editing.

5) Fig. 5: include legend

6) The calibration procedure has been described in detail, but an English language check is required. The text partly reads like copied from a calibration manual, e.g. “Then taking a soil sample using a cutting ring (100 cm3 in volume) at its adjacent position and weighing its fresh weight. After that, put the remaining soil back into the container used for mixing. Then repeating all the procedures above until the soil moisture content was saturated.”

7) Thiessen polygon (TP) approach: I miss a strong argumentation for using the TP approach instead of a simple average – especially since using the average does not lead to significantly different results (as shown in the appendix). Please elaborate on this.

8) line 291 “divided by the AMSR2 pixel and the FY3B pixel” should be “separately for the AMSR2 and FY3B pixels”

9) Fig. 10: Do you have any idea why the rainfall event on DOY 177 is not reflected more clearly in the SM time series?

10) line 306: “That indicates that … the up-scaled average value could well represent the average SM of the entire pixel”. So why don’t you just use the average of all SM stations in one pixel? (see also point 7)

11) line 308: “The results of the TP method were mainly utilized...” - “mainly” suggests that other methods were used too, but to my understanding you used TP exclusively. Is that correct?

12) Fig. 11: (d) and (e) should be switched. Figure layout should be improved. This figure shows again the similarity between the TP method and a simple average. Please make it very clear in your methods section why you decided to use TP.

13) Fig. 12: These 6 figures can be reduced to one: Plot all SM stations in one figure (as there is not much difference anyway between the stations located in the AMSR2 and FY3B pixels), and use two different colors to account for measurements during vegetation stage 1 and 2.

14) “3.2 Evaluation with In Situ Data and Inter-Comparison of Each Product” - consider naming this section “Satellite data evaluation”, to make clear what is the different to section 3.1

15) Fig. 13: If there is so little data available for FY3B, does it even make sense to include them in the study? What is the reason for such a big data gap?

16) Fig. 14: Again, those 9 figures can be reduced to only 3, if different colors for stage 1 and 2 are used

17) Please elaborate on how meaningful these metrics are, as they are based on only 23 days of data. Include significance levels for the correlation coefficient.

18) line 436: be careful with statements like “the FY3B product performs better than the JAXA and LPRM products at this time”, if the analyzed period is so short (and different for the satellite products), and no significance metrics have been calculated.

19) Can the large bias between LPRM and in situ be explained somehow, e.g. by looking at other studies that have evaluated AMSR2 SM?

20) Fig. 16: Again, all three columns can be combined in one plot by using two colors for stage 1 and stage 2. In the accompanying text, the figures are only described, but no interpretation is provided. Why is e.g. the relationship JAXA-SST positive, but LPRM-SST negative? Moreover, high SM values in JAXA (stage 2) can probably be explained by rainfall events. So, how meaningful is a comparison to SST?

21) Fig. 17: Again, all (sub)figures should share the same layout. Moreover, it is not mentioned in the caption why there are three columns (a-f). Figures + captions should be self-explaining – mention again that the columns show the entire period/stage 1/stage 2 (or reduce to 1 column by using different colors for stage 1 and 2)

22) Appendix: To my understanding, these tables show again that the TP method is not superior to using a simple average.

This manuscript presents a nice validation study of different satellite products. I appreciate the details provided on the calibration etc. of the network installation, but I miss a more comprehensive discussion and interpretation of the obtained validation metrics. Right now, the discussion section contains new results (scatter plots with EVI, SST and in situ SM), but not a lot of additional interpretation, which would make the entire manuscript more interesting and valuable to other scientists in the community. Especially, I would expect a section on how meaningful the obtained results are regarding the short study period of only a few months of the same year. The authors do not relate their results to results found in other, similar validation studies. Almost all figures should be carefully re-worked (standardized layout, correct/complement captions, usage of colors in order to reduce number of subplots).

Author Response

Dear Reviewer,

Firstly, we are so sorry for the late submission of our manuscript. Thank you very much for your comments concerning our manuscript. Those comments are all valuable and very helpful for revising and improving our paper, as well as the important guiding significance to our researches. We have modified each suggestion accordingly, and I believe the article has been comprehensively improved. We very much hope that our revision could satisfy you. If there are any other mistakes or improper writing, please inform us and we will modify carefully. Finally, we apologize again for the late submission and thanks for your work. Look forward to hearing from you soon.

The responses are as follows:

Point 1: Abstract: you describe what you did and what the resulting metrics are, but you should include at least 1-2 sentences of interpretation of those results. Sentences like “The FY3B product is seriously missing” are confusing, if the reader is not familiar with the data set and the rest of the manuscript.

Response 1: Thanks for pointing out the shortcomings in the abstract, I have added some explanations in the text and rewritten the sentences about the FY3B.

Point 2: Introduction – last paragraph: formulation of distinct science questions would help (i.e., why was the performance of the products discussed according to vegetation, SST and actual SM).

Response 2: Thank you for reminding me of that. I’ve added some description at the last paragraph in the introduction.

Point 3: Fig. 1b seems to be wrong – the caption refers to Fig. 2. An idea would be to keep Fig. 1b but instead of the relatively homogeneous (and therefore not very interesting) Landsat image display land cover in the background. Why is the SMAP pixel shown in Figs. 1 and 2?

Response 3: Yes! Thank you for pointing out the mistake!

The observation network was established with the SMAP pixel as spatial reference, so the SMAP pixel is shown in the figure. I’ve already added more details about it in Section 2.1 to make readers understand it.

Point 4: Fig. 4, but relevant throughout the manuscript: Using the same layout and design style for all figures would make the manuscript look more professional. Right now, it looks as if a number of people have worked on the paper who just put their results together without any final layout editing.

Response 4: The figure styles were actually not uniform and not beautiful, I have re-created them.

Point 5: Fig. 5: include legend

Response 5: Different colours in Fig. 5 only represent different sensors in the calibration test. There is no more meanings but to distinguish them. Therefore, the legend is not used so as not to make the figures look very complicated. Instead, they are described in the caption.

Point 6: The calibration procedure has been described in detail, but an English language check is required. The text partly reads like copied from a calibration manual, e.g. “Then taking a soil sample using a cutting ring (100 cm3 in volume) at its adjacent position and weighing its fresh weight. After that, put the remaining soil back into the container used for mixing. Then repeating all the procedures above until the soil moisture content was saturated.”

Response 6: Thank you for pointing out the details. I have noticed that and rewritten the sentences. Hope them improved.

Point 7: Thiessen polygon (TP) approach: I miss a strong argumentation for using the TP approach instead of a simple average – especially since using the average does not lead to significantly different results (as shown in the appendix). Please elaborate on this.

Response 7: The TP approach as an up-scaled method takes the spatial distribution of in situ points into account, while direct average treats the proportion of each point equally. The difference between them is small when the distribution of the points is uniform. But when some points are gathered together and they have a significant spatial difference from other locations in the pixel, the direct average will be overestimated or underestimated because of the high or low data of the gathered points. But it could be avoided by the TP approach. These above have also been described in Section 2.5.1.

Point 8: line 291 “divided by the AMSR2 pixel and the FY3B pixel” should be “separately for the AMSR2 and FY3B pixels”

Response 8: Thank you for pointing out the detail! I’ve corrected it.

Point 9: Fig. 10: Do you have any idea why the rainfall event on DOY 177 is not reflected more clearly in the SM time series?

Response 9: After re-examining the precipitation data, it was found that the time was wrong because of confusing the GMT and local time. But the problem was still existing after the correction. All times in this article are GMT, except as noted separately.

The precipitation data from the meteorological station is the cumulative amount every 3 hours. The daily precipitation data in this paper was the accumulated precipitation data from the past 24 hours since the satellite transited. The large amount of precipitation that occurred in 176th day was mainly concentrated at 3 to 6 o’clock, which was a short-term concentrated precipitation. The satellite transit time was around 17:30, with a gap of 9.5-12.5 hours. The in situ data showed that most of the in situ points barely reacted. The meteorological station is about 60km away from the study area, and the study area covered 36 km×36 km. It may be that the precipitation in the area was not obvious.

On the other hand, the EVI rose sharply and the crop grew rapidly at this time. The surface temperature was also at the highest value during the experimental period. So the evaporation and transpiration cannot be ignored. The moisture might be reduced largely before the satellite transited. Therefore, for all the analysis above, the high precipitation at 176th day did not lead to an increase in soil moisture.

Point 10: line 306: “That indicates that … the up-scaled average value could well represent the average SM of the entire pixel”. So why don’t you just use the average of all SM stations in one pixel? (see also point 7)

Response 10: In this experiment, we usually placed more than one sensor in close distance to prevent the individual sensor from failing, and most of the results of the TP approach and the direct average were also very close. To some extent, that indicated that the in situ points were evenly distributed and the spatial variation within the pixel was stable and uniform. However, there were also some large differences between the results of the TP approach and the direct average because of the big changes of the gathered points’ data. Because such situations were relatively small, the impacts on the whole time period were limited, so the statistical results also would not be significantly affected. In order to eliminate the error caused by the artificial selection of points, the TP results were preferentially utilized for calculation. The direct average results were also given as a reference in the attached table.

These above have also described in Section 3.1.

Point 11: line 308: “The results of the TP method were mainly utilized...” - “mainly” suggests that other methods were used too, but to my understanding you used TP exclusively. Is that correct?

Response 11: When choosing the averaging methods, we also try other differential methods such as Kriging interpolation. But the calculations and operation of the other methods were not convenient. And there were no better results. We also did not understand the other methods totally. So the other methods and results were not explained and used in this paper. But the “mainly” in the text did seem confusing. I decide to delete it as no other methods are shown in the paper.

Point 12: Fig. 11: (d) and (e) should be switched. Figure layout should be improved. This figure shows again the similarity between the TP method and a simple average. Please make it very clear in your methods section why you decided to use TP.

Response 12: Yes! I’ve switched them, and the figures have been improved. More details about TP, please see Response 7, 10 and 11.

Point 13: Fig. 12: These 6 figures can be reduced to one: Plot all SM stations in one figure (as there is not much difference anyway between the stations located in the AMSR2 and FY3B pixels), and use two different colors to account for measurements during vegetation stage 1 and 2.

Response 13: I’ve modified the figures and use different colors for different stages. Hope them improved!

Point 14: 3.2 Evaluation with In Situ Data and Inter-Comparison of Each Product” - consider naming this section “Satellite data evaluation”, to make clear what is the different to section 3.1

Response 14: Thank you for pointing out the detail! I’ve corrected it.

Point 15: Fig. 13: If there is so little data available for FY3B, does it even make sense to include them in the study? What is the reason for such a big data gap?

Response 15: We want to understand the advantages and disadvantages of the respective algorithms and platforms by comparing different X-band soil moisture products in this paper. As described in the paper, the brightness temperature data of the FY3B is good, and it is consistent with the AMSR2. Therefore, it can be judged that the loss soil moisture data is mainly caused by the soil moisture retrieval algorithm of the FY3B. Therefore, the research on FY3B soil moisture can focus on the applicability and improvement of the retrieval algorithm. The Global surface soil moisture monitoring using X-band has been over a very long time. There was a few months before the AMSR2 operating as the successor of AMSR-E. The FY3B also has X-band and the same equatorial local crossing time as the AMSR2. Therefore, the research and improvement of the FY3B soil moisture products will be of great significance to the consistent continuity of the application of X-band to global soil moisture monitoring.

Point 16: Fig. 14: Again, those 9 figures can be reduced to only 3, if different colors for stage 1 and 2 are used.

Response 16: The figures have been reduced to three and different colors were used to indicate the different stages.

Point 17: Please elaborate on how meaningful these metrics are, as they are based on only 23 days of data. Include significance levels for the correlation coefficient.

Response 17: The FY3B brightness temperature of has always been stably obtained and consistent with the AMSR2, while the FY3B soil moisture product is beyond the saturation for most of the experimental period. Therefore, the FY3B product cannot be evaluated throughout the whole experimental period. And we hope to evaluate and compare the limited data of the FY3B product under the saturation and to know whether it is worth improving. Therefore, the three products were evaluated as the presence of the FY3B product. In the results, only the LPRM product has a high correlation coefficient, and its significance level has also been given in Section 3.2.

Point 18: line 436: be careful with statements like “the FY3B product performs better than the JAXA and LPRM products at this time”, if the analyzed period is so short (and different for the satellite products), and no significance metrics have been calculated.

Response 18: Yes! We’ve noticed that. The expression has been improved. And the differences between the three products and the in situ data were analysed after the recovery of the FY3B product in the later experimental period. It was found that the performance of FY3B seems to be better than the JAXA and LPRM products. The P values in the significance level were 0.0341 and 0.0001 respectively.

These above are also described in Section 4.1.

Point 19: Can the large bias between LPRM and in situ be explained somehow, e.g. by looking at other studies that have evaluated AMSR2 SM?

Response 19: By analysing the previous researches on the LPRM algorithm and evaluations and the results in this experiment, the large bias between the LPRM and in situ are explained. The details are in sections 3.2 and 4.3.

Point 20: Fig. 16: Again, all three columns can be combined in one plot by using two colors for stage 1 and stage 2. In the accompanying text, the figures are only described, but no interpretation is provided. Why is e.g. the relationship JAXA-SST positive, but LPRM-SST negative? Moreover, high SM values in JAXA (stage 2) can probably be explained by rainfall events. So, how meaningful is a comparison to SST?

Response 20: All the figures have been reworked to better explain the impact of the SST on results. The interpretation and details have been supplemented in the text. The meteorological station providing precipitation data is about 60km away from the study area. The soil moisture is obviously affected by precipitation events, but the increase of it is not linearly related to the amount of precipitation. Please see Response 9 for more details on this. The SST was in situ measurement like soil moisture, and both were consistent in time and space, which means that the SST is more accurate and reliable. On the other hand, the SST can directly affect the brightness temperature and emissivity of the surface soil, which will affect the retrieval results of soil moisture. Therefore, it is meaningful to study the effects of the SST on the results. These above were also explained in Section 4.2.

Point 21: Fig. 17: Again, all (sub)figures should share the same layout. Moreover, it is not mentioned in the caption why there are three columns (a-f). Figures + captions should be self-explaining – mention again that the columns show the entire period/stage 1/stage 2 (or reduce to 1 column by using different colors for stage 1 and 2)

Response 21: The figure has been modified and different colors were used to represent the 2 stage, so that it can be more clearly expressed.

Point 22: Appendix: To my understanding, these tables show again that the TP method is not superior to using a simple average.

Response 22: Please see Response 7 and 10.

Point 23: Especially, I would expect a section on how meaningful the obtained results are regarding the short study period of only a few months of the same year.

Response 23: Thank you for give me the suggestion. I have explained the meaning of this experiment before Section 4.1 in Chapter 4 as the suggestion. Due to its limited amount of text, it is not listed separately as a section.

Reviewer 2 Report

Dear Authors,

            I think that the your manuscript entitled  “Evaluation and Analysis of AMSR2 and FY3B Soil 2 Moisture Products by an In Situ Network in Cropland 3 on Pixel Scale in the Northeast of China” brings new and valuable ideas and results to agriculture soil moisture studies.

Especially important are detailed analyses related to an in situ soil moisture observation network at pixel scale which was constructed in cropland on pixel scale. 

This work is interesting from both scientific and practical point of view, and is suitable for publication in Remote Sensing, just major revision.

Remarks

1.      It seems to me that authors missed in bibliography some similar but important ground soil moisture studies for agriculture applicationsshowing recent achievements in this field, related specifically to remote sensing observations made in radiometric spectrum. This is clearly visible, for advanced reader in soil moisture studies trough whole paper, that references are rather domestic or too general (it can involve the publication bias). I understand that Authors focused rather on the JAXA and LPRM as well as on ANSR2 soil moisture assessments, although SMAP was twice mentioned on the literatureHowever, information about Europe’s SMOS satellite studies at pixel scale (also for  agriculture applications) were probably missed by the Authors. I would suggest to put (in line [32] together with references [1-6]) also another reference: SMOS data as a source of the agricultural drought information: Case study of the Vistula catchment, Poland, M Kędzior, J Zawadzki, Geoderma 306, 167-182, 2017.

2.      In general quality of figures should be substantially revised before accepting the paper for publication in scientific paper. Examples:

·         There are not units on Y-axis in Figure 4 and on abscissa in Figure 6 (I would suggest also to change colours in Figure 4 to make it more similar to the other figures. The same about points in Figure 4.

·         In Figure 11 the graph at the centre (b) and (e ) are located unevenly.

·         The trends in Figure 12 (or lack of them) would be rather unclear for the potential reader. Could you draw  them in the Figure 12 ?, to make description of the figure  (lines 328-348) much clearer. Improve the caption of the figure. What means “Among them? Use shorter sentences, but more precise, please.

·         Similar remarks to figures/captions of 14,15,16, and 17 (!). To be consistent remove y=x in Figure 17. Show R2 on each scatterplot.

·         In Figure 13 one Y-axis scale for precipitation is missed.

·         I would generalise that the quality of mentioned figures and their captions should be much better. I would even suggest to remove Figure 15 (or may be also) another. It is really necessary to show so many simple scatter plots?

3.      Check the English again focusing on errors and clarity. Many sentences are not clear enough. Just example (line 485-486) “However, the LPRM product does not have such a clear edge. Its change is more severe and the range is larger, 0~1 cm3/cm3, and so is its error.”           (In fact the edges are formed by the points of scatterplots, not by the LPRM product itself”, etc.

Or, (something wrong) (lines 478-479) “It can more intuitively reflect the impact of the actual surface soil moisture itself on the products’ performance than Figure 14”. There is really many such unclear or mistaken fragments through the manuscript, they cannot stay in scientific paper, so the paper should be substantially improved by Native Speaker, although the results obtained are quite interesting.

I recommend the manuscript for publication in Remote Sensing, after above-mentioned major revision.

                                                                                                                                                                                                                                                                                                                                        Sincerely yours

Reviewer

Author Response

Dear Reviewer,

Firstly, we are so sorry for the late submission of our manuscript. Thank you very much for your comments concerning our manuscript. Those comments are all valuable and very helpful for revising and improving our paper, as well as the important guiding significance to our researches. We have modified each suggestion accordingly, and I believe the article has been comprehensively improved. We very much hope that our revision could satisfy you. If there are any other mistakes or improper writing, please inform us and we will modify carefully. Finally, we apologize again for the late submission and thanks for your work. Look forward to hearing from you soon.

The responses are as follows:

Point 1: It seems to me that authors missed in bibliography some similar but important ground soil moisture studies for agriculture applications, showing recent achievements in this field, related specifically to remote sensing observations made in radiometric spectrum. This is clearly visible, for advanced reader in soil moisture studies trough whole paper, that references are rather domestic or too general (it can involve the publication bias). I understand that Authors focused rather on the JAXA and LPRM as well as on ANSR2 soil moisture assessments, although SMAP was twice mentioned on the literature. However, information about Europe’s SMOS satellite studies at pixel scale (also for  agriculture applications) were probably missed by the Authors. I would suggest to put (in line [32] together with references [1-6]) also another reference: SMOS data as a source of the agricultural drought information: Case study of the Vistula catchment, Poland, M Kędzior, J Zawadzki, Geoderma 306, 167-182, 2017.

Response 1: Due to the lack of understanding of the L-band retrieval of soil moisture, I am also looking for some good researches in this area. Thank you very much for providing me with such a very good choice. This article will be of great help to my understanding of the current research situation and my future research. I will quote it in my manuscript.

Point 2: In general quality of figures should be substantially revised before accepting the paper for publication in scientific paper. Examples:

There are not units on Y-axis in Figure 4 and on abscissa in Figure 6 (I would suggest also to change colours in Figure 4 to make it more similar to the other figures. The same about points in Figure 4.

In Figure 11 the graph at the centre (b) and (e ) are located unevenly.

The trends in Figure 12 (or lack of them) would be rather unclear for the potential reader. Could you draw them in the Figure 12 ?, to make description of the figure  (lines 328-348) much clearer. Improve the caption of the figure. What means “Among them? Use shorter sentences, but more precise, please.

Similar remarks to figures/captions of 14,15,16, and 17 (!). To be consistent remove y=x in Figure 17. Show R2 on each scatterplot.

In Figure 13 one Y-axis scale for precipitation is missed.

I would generalise that the quality of mentioned figures and their captions should be much better. I would even suggest to remove Figure 15 (or may be also) another. It is really necessary to show so many simple scatter plots?

Response 2:

Thanks for the careful attention to the details of my article. I have revised most of the figures and the captions and explanations of them to make them clearer and simpler for readers. Since the data in each scatter plot did not show a good linear relationship, they could not be well explained by the R2. And there was no good explanation for R2 itself in this study. So the R2 was not listed in order to avoid making the readers confused.

The precipitation coordinate was placed on the right side of the graph. Since it was consistent with the numerical range of temperature, they shared one Y-axis scale.

I have made some modifications to the figures by combining some redundant ones and representing different experimental periods in different colours. Due to the particularity of the study area, the experimental period is short. If the result were only expressed through tables, we were afraid that the experimental results cannot be fully expressed and some valuable laws might be missed. Therefore, we want to display the experimental results data as comprehensively as possible, and hope to have some inspirations to the readers. It is best to find some laws that we have not found.

Point 3: Check the English again focusing on errors and clarity. Many sentences are not clear enough. Just example (line 485-486) “However, the LPRM product does not have such a clear edge. Its change is more severe and the range is larger, 0~1 cm3/cm3, and so is its error.”           (In fact the edges are formed by the points of scatterplots, not by the LPRM product itself”, etc.

Or, (something wrong) (lines 478-479) “It can more intuitively reflect the impact of the actual surface soil moisture itself on the products’ performance than Figure 14”. There is really many such unclear or mistaken fragments through the manuscript, they cannot stay in scientific paper, so the paper should be substantially improved by Native Speaker, although the results obtained are quite interesting.

Response 3: Thank you for picking up my unclear sentences! I have modified them and hope that the results of the modifications will meet the requirements.

Reviewer 3 Report

This paper reports interesting comparison between AMSR2 SM data (both the products generated respectively by NASA and JAXA) and the FY3B SM products with ground data collected in situ in cropland in the Northeast of China.

The first suggestion is to insert some example of inversion models of surface SM and temperature (row 40).

The second suggestion is to insert in the introduction a little nod to ESA SMOS mission and NASA SMAP mission even if they operate at different band respect to the sensors used in this context.(rows 54-56).

In some sections the speech is too long and the risk is to divert the reader from the aim of the paper. 

I propose minor revision.

Author Response

Dear Reviewer,

Firstly, we are so sorry for the late submission of our manuscript. Thank you very much for your comments concerning our manuscript. Those comments are all valuable and very helpful for revising and improving our paper, as well as the important guiding significance to our researches. We have modified each suggestion accordingly, and I believe the article has been comprehensively improved. We very much hope that our revision could satisfy you. If there are any other mistakes or improper writing, please inform us and we will modify carefully. Finally, we apologize again for the late submission and thanks for your work. Look forward to hearing from you soon.

The responses are as follows:

Point 1: The first suggestion is to insert some example of inversion models of surface SM and temperature (row 40).

Response 1: Thank you for reminding me of that. I’ve already added some references as examples of inversion models of surface SM and temperature.

Point 2: The second suggestion is to insert in the introduction a little nod to ESA SMOS mission and NASA SMAP mission even if they operate at different band respect to the sensors used in this context.

Response 2: I have added some descriptions about the ESA SMOS mission and the NASA SMAP mission in the introduction.

Point 3: In some sections the speech is too long and the risk is to divert the reader from the aim of the paper.

Response 3: Thank you for pointing out my speech problems! I have modified some difficult speeches in some sections and hope that the results of the modifications will meet the requirements.

Round 2

Reviewer 1 Report

Dear authors,

thank you for taking the time for revising your manuscript so carefully. I am fully satisfied with your answers to my questions and comments, and especially pleased with the improvement of the overall appearance of your manuscript (e.g. due to the improved figures).

Reviewer 2 Report

Dear Authors

Thank you for your careful revision of the manuscript entitled “Evaluation and Analysis of AMSR2 and FY3B Soil 2 Moisture Products by an In Situ Network in Cropland 3 on Pixel Scale in the Northeast of China”, for answers to my questions, as well as for clarification of the problems I mentioned. I am fully satisfied, knowing how complicated and difficult are such systematic remote sensing analyses combining soil moisture, climate and  vegetation. 

               I appreciate both scientific and practical value of the paper.

Sincerely Yours,

                                                                                                                                                                                                                                                                          Reviewer